

# Improving snow albedo modeling in E3SM land model (version 2.0) and assessing its impacts on snow and surface fluxes over the Tibetan Plateau

Dalei Hao[1], Gautam Bisht[1], Cenlin He[2], Edward Bair[3], Huilin Huang[1], Cheng Dang[4], Karl Rittger[5], Yu Gu[6], Hailong Wang[1], Yun Qian[1] and L. Ruby Leung[1]

[1]Atmospheric Sciences and Global Change Division, Pacific Northwest National Laboratory, Richland, WA, USA
[2]Research Applications Laboratory, National Center for Atmospheric Research, Boulder, CO, USA
[3]Earth Research Institute, University of California, Santa Barbara, CA, USA
[4]Joint Center for Satellite Data Assimilation, University Corporation for Atmospheric Research, Boulder, CO, USA
[5]Institute for Arctic and Alpine Research, University of Colorado, Boulder, CO, USA
[6]Joint Institute for Regional Earth System Science and Engineering and Department of Atmospheric and Oceanic Sciences, University of California, Los Angeles, CA, USA

*Correspondence to*: Dalei Hao (dalei.hao@pnnl.gov)

**Abstract.** With the highest albedo of the land surface, snow plays a vital role in Earth's surface energy and water cycles. Snow albedo is greatly affected by snow grain properties (e.g., size and shape) and light absorbing particles (LAPs) such as black carbon (BC) and dust. The mixing state of LAPs in snow also has large impacts on LAP-induced snow albedo reduction and surface radiative forcing (RF). However, most land surface models assume that snow grain shape is spherical and LAPs are externally mixed with the snow grains. This study improves the snow radiative transfer model in the land model (ELM v2.0) of the Energy Exascale Earth System Model version 2.0 (E3SM v2.0) by considering non-spherical snow grain shapes (i.e., spheroid, hexagonal plate and Koch snowflake) and internal mixing of dust-snow and systematically evaluates the impacts on surface energy and water balances over the Tibetan Plateau (TP). A series of ELM simulations with different treatments of snow grain shape, mixing state of BC-snow and dust-snow, and sub-grid topographic effects (TOP) on solar radiation are performed. Compared with two remote sensing snow products derived from the Moderate Resolution Imaging Spectroradiometer, the control ELM simulation with the default settings of spherical snow grain shape, internal mixing of BC-snow, external mixing of dust-snow and without TOP can capture the overall snow distribution reasonably. The estimated LAP-induced RF ranging from 0 to 21.9 W/m$^2$ with the area-weighted average value of 1.3 W/m$^2$ is comparable to reported values. Focusing on the snow-related processes and surface energy and water balances, Koch snowflake shape, among other non-spherical shapes, shows the largest difference from spherical shape in spring. The impacts of the mixing state of LAP-snow are smaller than the shape effects and depend on snow grain shape. Compared to external mixing, internal mixing of LAP-snow can lead to larger snow albedo reduction and snowmelt, which further affect surface energy and water cycles. Compared to the control simulation, the individual contributions of non-spherical snow shape, mixing state of LAP-snow, and local topography to the change of snow and surface fluxes have different signs and magnitudes, and their combined effects



may be negative or positive due to complex and non-linear interactions among the factors. Overall, the changes of net solar radiation in spring due to individual and combined effects range from -28.6 to 16.9 W/m$^2$ and -29.7 to 12.2 W/m$^2$, respectively. This study advances understanding of the role of snow grain shape and mixing state of LAP-snow in land surface processes and offers guidance for improving snow simulations and RF estimates in Earth system models under climate change.






## 1 Introduction

With the highest albedo of all land surfaces, snow plays an important role in Earth's radiation budget (Flanner et al., 2011), water cycle (Barnett et al., 2005), and regional/global climate change via positive snow-albedo feedback (Hall, 2004; Riihelä et al., 2021). Snow albedo ($\alpha_{sno}$) is regulated by solar zenith angle (SZA), sky conditions, snow properties, and snow impurities

such as light absorbing particles (LAPs) (He and Flanner, 2020). Snow properties such as snow grain size and shape have large influences on single scattering optical properties of snow (Dang et al., 2016; Tanikawa et al., 2020; Warren, 1982). LAPs that include black carbon (BC), light-absorbing organic carbon (also known as brown carbon, BrC), and mineral dust can darken the snow-covered surface (Wu et al., 2018; Zhao et al., 2014), increasing the amount of energy absorbed by snow and accelerating snowmelt (Kang et al., 2020; Sarangi et al., 2019). The surface radiative forcing (RF, a measure of the change in

surface radiative flux) from LAPs has contributed to rapid glacier retreat (Xu et al., 2009). BC is the optically absorbing component of soot and has been identified as one of the most important anthropogenic emissions affecting global climate due to its large RF (Bond et al., 2013; Flanner et al., 2007; Hadley and Kirchstetter, 2012). BrC absorbs more light at shorter wavelengths compared to BC (Andreae and Gelencsér, 2006). The impurity effects of BrC in snow on modelling $\alpha_{sno}$ and its RF have not been widely investigated due to its large variabilities and uncertainties in the optical properties (Brown et al.,

2021; Skiles et al., 2018). Dust has smaller mass absorption efficiency compared to BC, but the impurity effects of dust on snow albedo reduction (SAR) and its RF are found to be comparable to or even greater than BC in the high-mountain Asia (HMA) and the Sierra Nevada due to the larger dust deposition (Kaspari et al., 2014; Sarangi et al., 2020; Sterle et al., 2013; Usha et al., 2020).

Considering the high sensitivity of $\alpha_{sno}$ to snow grain shape, the assumptions of spherical snow grain shape may be inappropriate to realistically model $\alpha_{sno}$. Radiative transfer models or snow albedo parameterizations in the Earth System Models (ESMs), e.g., the Snow, Ice, and Aerosol Radiative (SNICAR) model (Flanner et al., 2007), well explain the dependence of $\alpha_{sno}$ on snow grain size, snow depth, SZA, underlying surface, sky conditions, and spectral wavelength. However, the snow grain shape in these models/parameterizations is generally assumed to be sphere. In reality, snow grain is

usually irregular and non-spherical (Kokhanovsky et al., 2005), and depends on snow age and meteorological conditions (Dominé et al., 2003; LaChapelle, 1969). Compared to spherical grains, non-spherical snow grains show weaker forward scattering and thus smaller asymmetry factor, resulting in higher albedo (Dang et al., 2016; Libois et al., 2013; Räisänen et al., 2015; Räisänen et al., 2017). He et al. (2017, 2018b) developed parameterizations to explicitly represent the impacts of three typical non-spherical grain shapes (i.e., spheroid, hexagonal plate and Koch snowflake) on single-scattering properties of snow

based on the geometric-optics surface wave (GOS) approach. These parameterizations have been implemented in the standalone version of the SNICAR model (He et al., 2018a) and have further been incorporated into the latest SNICAR with Adding-Doubling solver, version 3.0 (SNICAR-AD v3) (Flanner et al., 2021). However, these new parameterizations have not been incorporated into ESMs and applied at regional or global scales yet.





The mixing of LAPs with snow via dry deposition (external mixing) and wet deposition (internal mixing) has large impacts
     on $\alpha_{sno}$, and only accounting for external mixing of LAP-snow could lead to underestimations of modeled SAR and RFs
     (Flanner et al., 2012; Liou et al., 2014). LAP-snow internal mixing can also occur through dry deposition followed by
     successive snow events over high elevation regions (Liou et al., 2014). Compared to external mixing, internal mixing of LAP-
     snow can enhance SAR and thus lead to larger RF (Flanner et al., 2012; He et al., 2018b; Liou et al., 2014; Tanikawa et al.,
2020). However, snow albedo models/parameterizations in ESMs often assume that LAPs are only externally mixed with snow
     (Flanner et al., 2007). Flanner et al. (2012) showed that the internal mixing of BC-snow for spherical snow grain shape can be
     accurately described by the dynamic effective medium approximation (DEMA) (Chýlek and Srivastava, 1983). He et al. (2017,
     2018a, 2018b) implemented a series of computationally efficient polynomial-based parameterizations to represent the effects
     of BC-snow internal mixing on $\alpha_{sno}$ based on GOS simulations for both spherical and non-spherical snow grain shapes. He et
al. (2019) further developed parameterizations to quantify the effects of dust-snow internal mixing for different snow grain
     shapes. These recent progresses provide a good opportunity to account for the LAP-snow internal mixing in ESMs and quantify
     their effects on land surface processes.

     Realistically representing the impacts of snow grain properties and mixing state of LAPs on $\alpha_{sno}$ in ESMs is required for
improving the simulations of snow processes and surface energy and water budgets. However, the non-spherical grain shape
     and internal mixing of dust-snow are still not represented in the snow radiative transfer model of the land model (ELM v2.0)
     of the Energy Exascale Earth System Model version 2.0 (E3SM v2.0) . E3SM, supported by the U.S. Department of Energy
     (DOE), is a state-of-the-science fully coupled ESM aimed at improving projecting future changes of the water cycle,
     biogeochemistry, and cryosphere systems (Golaz et al., 2019; Leung et al., 2020). As the land component of E3SM v2.0, ELM
v2.0 was originally developed from the Community land model version 4.5 (CLM4.5), which uses SNICAR to model the snow
     radiative transfer processes. Dang et al. (2019) incorporated a two-stream radiative transfer scheme with delta-Eddington
     approximation and adding–doubling technique into SNICAR of ELM v2.0 (named as SNICAR-AD). This new SNICAR-AD
     model improves and unifies the treatment of snow shortwave radiative transfer in the land and sea ice components of E3SM
     (Dang et al., 2019). The internal mixing of BC-snow has been incorporated in SNICAR-AD of ELM v2.0 using a new look-
up table derived from the DEMA approach (Wang et al., 2020a). However, the non-spherical snow grain shape and internal
     mixing of dust-snow are not included in the SNICAR-AD of ELM v2.0. The impact of non-spherical grain shape and internal
     mixing of LAP-snow on $\alpha_{sno}$ has been well studied using the snow albedo models that haven't been coupled to land surface
     model (Dang et al., 2016; He et al., 2018a; Räisänen et al., 2015). Inclusion of parameterizations for non-spherical grain shape
     and LAP-snow internal mixing into the snow radiative transfer model of ELM v2.0 offers a feasible way to analyze the
sensitivity of surface energy and water budgets to such changes and quantify the corresponding uncertainties. Moreover, an
     improved parameterization of sub-grid topographic effects on solar radiation has been recently implemented in ELM v1.0 (Hao



et al., 2021a), which makes it possible to study the interactions between improved snow radiative transfer model and sub-grid topographic effects on solar radiation.

This study aims to improve the SNICAR-AD in ELM v2.0 by accounting for non-spherical snow grain shape and internal mixing of LAP-snow and systematically evaluate their impacts on surface energy balance and water cycles over the Tibetan Plateau. The analytical parameterizations of He et al. (2017, 2018a, 2018b) for three non-spherical grain shape (i.e., spheroid, hexagonal plate and Koch snowflake) and He et al. (2019) for internal mixing of dust-snow are implemented in SNICAR-AD of ELM v2.0. A series of ELM simulations with different assumptions of snow grain shape and mixing state of LAP-snow are

carried out. Then a control ELM simulation with the default settings is compared with two remote sensing products derived from the Moderate Resolution Imaging Spectroradiometer (MODIS). The RFs from LAPs' impurity effects in the control experiment is also compared to the values reported in existing studies. The impacts of snow grain shape and mixing state of LAP-snow on the snow-related processes, energy cycle, and water cycle are analyzed. The interactions between the improved snow radiative transfer model and sub-grid topographic effects on solar radiation are also investigated.

**2 Model Improvements in ELM v2.0**

**2.1 Snow albedo modeling in ELM and SNICAR-AD v3**

E3SM v2.0 (including ELM v2.0) was released on September 29, 2021 (https://github.com/E3SM-Project/E3SM/releases/tag/v2.0.0). The default settings for hydrologic and biogeochemistry models in ELM v2.0 are the same as ELM v1.0 and ELM v1.1. ELM v1.0 incorporated new features for better representing soil hydrology and carbon cycle

dynamics (Golaz et al., 2019). New options for modeling $\alpha_{sno}$ are implemented in ELM v2.0. The SNICAR-AD model is also adopted in ELM v2.0 to more accurately describe the multi-layer snow radiative transfer processes (Dang et al., 2019). SNICAR-AD of ELM showed better performance in simulating $\alpha_{sno}$ for both visible (VIS) and near-infrared (NIR) bands at various SZAs under all sky conditions than the original SNICAR (Dang et al., 2019). The internal mixing of hydrophilic BC-snow is represented in SNICAR-AD of ELM v2.0 using a new look-up table derived using the DEMA approach (Wang et al.,

2020a). In parallel, Hao et al. (2021a) updated the two-stream radiative transfer scheme of ELM v1.0 to account for the sub-grid topographic effects on solar radiation (TOP), which showed improved performance on simulating surface energy balance and snow processes over the Tibetan Plateau (TP) compared to the original plane-parallel (PP) scheme.

SNICAR-AD v3 includes numerous recent improvements and extensions in SNICAR (Flanner et al., 2021). Specifically, it

adopts the adding-doubling solver for the radiative transfer equations; adds new representations of carbon dioxide snow, snow algae, and new types of dust, BrC, and volcanic ash; contains options for ice refractive indices; parameterizes the impacts of non-spherical snow grain shape; and considers the SZA dependence of surface spectral irradiances under different atmospheric profiles. However, internal mixing of LAP-snow is not accounted for in SNICAR-AD v3.





As part of this study, we implement the following new features into the snow albedo model in ELM v2.0: 1) SZA dependence of surface irradiance; 2) the parameterization of He et al. (2017, 2018a, 2018b) for non-spherical snow grain shape (i.e., spheroid, hexagonal plate, and Koch snowflake); and 3) the parametrization of He et al. (2019) for internal mixing of dust-snow. For the first enhancement, new look-up tables of SNICAR-AD v3 are incorporated in ELM v2.0, which considers the surface irradiance dependence on SZA under six types of atmospheric profiles: mid-latitude winter, mid-latitude summer, sub-

Arctic winter, sub-Arctic summer, summit Greenland, and high mountain. The remaining two enhancements are described in Sections 2.2 and 2.3.

## 2.2 Parameterizing the impacts of snow grain shape

Snow grain shape affects the single scattering optical properties, such as single-scattering albedo ($\omega$), extinction cross section ($Q_{ext}$), and asymmetry factor (g). Previous studies represented non-spherical grains as spheres of an equivalent radius, $R_e$, such

that the specific surface area (i.e., surface area to mass ratio) is equal to that of the non-spherical grain. Although the equivalent sphere performs well in computing $\omega$ and $Q_{ext}$ (Grenfell and Warren, 1999), it is not sufficiently accurate in estimating g (Dang et al., 2016). He et al. (2017) defined a specific-projected-area-equivalent radius, $R_s$, of an equivalent sphere with the same orientation-averaged projected area to volume ratio as that of the non-spherical grain:

$$R_s = \frac{3V_{snow}}{4A_{snow}} \tag{1}$$

where $V_{snow}$ is the volume of snow grain and $A_{snow}$ is the projected area of a snow grain averaged over all directions. For convex shapes (e.g., spheroid and hexagonal plate), $R_s = R_e$, whereas for concave shape (e.g. Koch snowflake), $R_s > R_e$ generally (He et al., 2017). He et al. (2017) developed new parameterizations to further consider the effects of non-spherical grain shape on g, by introducing two additional factors for describing the degree of non-sphericity: (1) aspect ratio (AR) which is the ratio of grain width to length, and (2) shape factor (SF) which is the ratio of $R_s$ of a non-spherical grain to that of an equal-volume

sphere. Specifically, three typical non-spherical grain shapes (Figure S1) are considered: spheroid, hexagonal plate and Koch snowflake, with their recommended AR and SF values listed in Table 1 of He et al. (2017).

According to (Fu, 2007), the asymmetry factor for hexagonal plate/column, $g_{hex}$, can be derived by

$$g_{hex} = \frac{1-g'}{2\omega} + g' \tag{2}$$

where $g'$ is as

$$g' = \begin{cases} a_0 + a_1 \cdot AR + a_2 \cdot AR^2, 0.1 \leq AR \leq 1.0 \\ b_0 + b_1 \cdot \ln(AR) + b_2 \cdot \ln^2(AR), 1 < AR \leq 20.0 \end{cases} \tag{3}$$

where both $a_i$ and $b_i$ are wavelength-dependent fitted coefficients provided in Tables 1 and 2 of Fu (2007). $\omega$ is the snow single-scattering albedo, and $AR$ is the snow grain aspect ratio. The asymmetry factor for other non-spherical shapes (i.e., spheroid and Koch snowflake), $g_{ns}$, can be calculated by

$$g_{ns} = g_{hex} \cdot C_g \tag{4}$$





where $C_g$, an empirical correction factor, is parameterized as the function of shape factor (SF) and $R_s$:

$$C_g = c_0 \cdot \left(\frac{SF_{ns}}{SF_{hex}}\right)^{c_1} \cdot (2R_s)^{c_2} \tag{5}$$

where $c_i$ are wavelength-dependent fitted coefficients provided in Table 3 of He et al. (2017), and Rs is the specific-projected-area-equivalent radius. These parameterizations have shown good performance when compared to the sophisticated geometric optics ray-tracing simulations (Flanner et al., 2021; He et al., 2017).

## 2.3 Parameterizing the impacts of internal mixing of dust-snow

The mixing state of dust-snow (Figure S1) mainly contributes to the change of ω (He et al., 2019; Liou et al., 2014). He et al. (2019) proposed a parameterization to account for the internal mixing of dust-snow by modifying ω. Specifically, compared to pure snow, the snow single-scatter co-albedo (1-ω) enhancement ($E_{1-\omega}$) caused by dust-snow internal mixing was defined as

$$E_{1-\omega} = \frac{1-\omega_{dust}}{1-\omega_p} \tag{6}$$

where $\omega_{dust}$ and $\omega_p$ are single-scattering albedo of pure snow and dirty snow internally mixed by snow. $E_{1-\omega}$ can be empirically calculated by

$$E_{1-\omega} = d_0 + d_1 \cdot (C_{dust})^{d_2} \tag{7}$$

where $C_{dust}$ is the dust mass concentration and $d_i$ with $i = 1,2$ are wavelength-dependent fitted coefficients listed in Table 1 of He et al. (2019). This parameterization has shown good consistencies with the Monte Carlo ray-tracing simulations (He et al., 2019).

## 3 Model Simulations and Remote Sensing Data

### 3.1 Experimental Design

Tibetan Plateau (TP), characterized by complex topography and frequent snow cover, is selected as the study area. Some regions of HMA include areas where LAPs have large impacts on snowmelt (Sarangi et al., 2020). This study area has a large elevation gradient ranging from below 1500 m to above 6000 m (Figure 1). Only model results for grid cells with elevation above 1500 m are considered in the study.

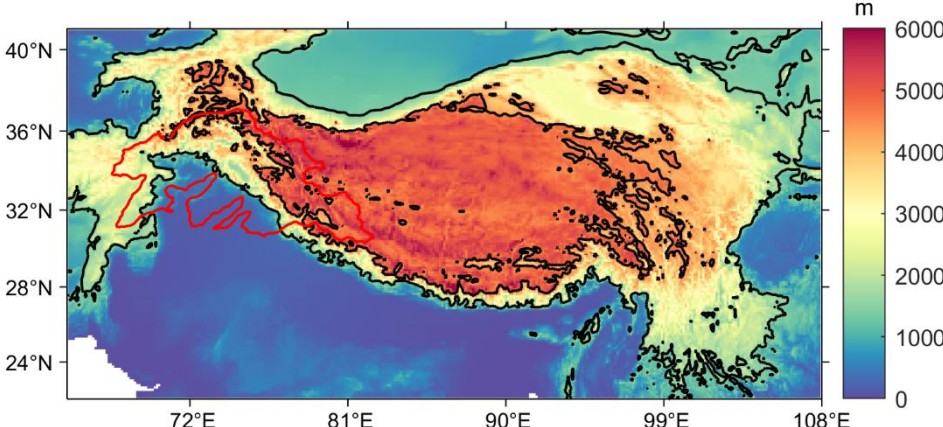

**Figure 1: Spatial distribution of elevation and the contour lines of 1.5 and 4.5 km over the Tibetan Plateau. The red line represents**
**the border of snowy regions of the Indus river basin acquired from the CHARIS project (doi.org/10.7265/TQFB-M828).**

A series of ten ELM simulations with different configurations at 0.125° spatial resolution are conducted to investigate the
sensitivity to the new improvements in the snow albedo model (Table 1). For all cases, the prescribed satellite phenology (SP)
mode is used, where the 8-day 500 m MODIS leaf area index (LAI) data is upscaled to derive the climatological monthly data
with 0.125° spatial resolution (Myneni et al., 2002). The Global Soil Wetness Project Phase 3 (GSWP3) v1 forcing data
(Dirmeyer et al., 2006) with 0.5° spatial resolution and 3-hourly temporal resolution is used to drive ELM. The prescribed
climatological monthly aerosol deposition data at 1.9°×2.5° is used from the Community Atmosphere Model version 5 coupled
with chemistry (Lamarque et al., 2010; Liu et al., 2012). The bilinear interpolation technique is used to spatially downscale
the GSWP3v1 and aerosol deposition data to 0.125° spatial resolution. The temporal downscaling of the solar data,
precipitation data, and all the other atmospheric data are performed using the cosine of the SZA-based, nearest-neighbour, and
linear interpolation methods, respectively. The configurations of the snow albedo model include (1) mid-latitude winter
atmosphere profile, (2) SZA-dependence of solar irradiance, (3) external mixing of hydrophobic BC with snow, (4) four
different snow grain shapes (i.e. sphere, spheroid, hexagonal plate, Koch snowflake), (5) internal and external mixing of
hydrophilic BC and dust with snow, and (6) two different solar radiation parameterizations (i.e., TOP and PP). The
parameterization of He et al. (2019) for the effects of internally-mixed dust in snow assumes no BC present in the snow. Thus,
we do not include a model configuration that simultaneously considered the internal mixing of BC and dust in snow. For each
case, ELM is first run from 1950 to 2000 for model spin-up, followed by another 10 years from 2001-2010 for model analysis.
The output variables are stored at a half-hourly interval to match the satellite observations and then aggregated to multi-year
averaged seasonal scales.



**Table 1. Model configurations with different snow grain shapes, mixing states of LAP-snow and solar radiation parameterizations.**

| Case ID | Snow Grain Shape | Mixing of hydrophilic BC-snow | Mixing of dust-snow | Solar radiation parameterization |
|---|---|---|---|---|
| Sph_BCInt_DExt_PP (Control) | Sphere | Internal | External | PP |
| Sphd_BCInt_DExt_PP | Spheroid | Internal | External | PP |
| Hex_BCInt_DExt_PP | Hexagonal Plate | Internal | External | PP |
| Koc_BCInt_DExt_PP | Koch Snowflake | Internal | External | PP |
| Sph_BCExt_DExt_PP | Sphere | External | External | PP |
| Sph_BCExt_DInt_PP | Sphere | External | Internal | PP |
| Koc_BCExt_DExt_PP | Koch Snowflake | External | External | PP |
| Koc_BCExt_DInt_PP | Koch Snowflake | External | Internal | PP |
| Sph_BCExt_DInt_TOP | Sphere | External | Internal | TOP |
| Koc_BCExt_DInt_TOP | Koch Snowflake | External | Internal | TOP |

## 3.2 MODIS data


Two snow products derived from MODIS onboard Terra are used to evaluate the ELM control simulations. The first set are the Snow-Covered Area and Grain Size (MODSCAG) and MODIS Dust and Radiative Forcing in Snow (MODDRFS) products (Painter et al., 2012; Painter et al., 2009). MODSCAG uses a physically-based spectral mixture algorithm to estimate the snow cover fractions ($f_{sno}$) and grain size and has been shown to have a better performance than MODIS MOD10A1 snow

cover products (Rittger et al., 2013). MODDRFS is used to calculate LAP-induced SAR based on the relative difference between satellite measured dirty $\alpha_{sno}$ and modeled pure $\alpha_{sno}$, which is derived from the grain size. The snow surface properties are further interpolated and smoothed to reduce the errors and uncertainties caused by off-nadir views, cloud contamination and data noise, resulting in improved quality and spatio-temporal consistency (Rittger et al., 2020) Snow albedo estimates combining MODSCAG and MODDRFS have a 4 to 6 percent RMSE and negligible bias (Bair et al., 2019). The

Snow Property Inversion from Remote Sensing (SPIReS) product is the second MODIS dataset used for evaluating ELM simulations. The SPIReS product builds on knowledge gained from MODSCAG and MODDRFS work and estimates LAP concentrations and snow grain size simultaneously with optimized data processing and integrated spatio-temporal interpolation and smoothing procedures (Bair et al., 2021c). Then LAP-induced SAR can be calculated by the albedo look-up table derived from a radiative transfer model (Bair et al., 2019). All the cloud and off-nadir effects, reflectance errors contribute to the

uncertainties in both snow products, and spherical snow grain shape is assumed in the MODSCAG/MODDRFS and SPIReS.

Both the daily 500 m $f_{sno}$ and SAR in MODSCAG and SPIReS products from 2001 to 2010 are used to compare with the ELM control simulation. SAR estimated by MODSCAG/MODDRFS covers the spectral range from 0.350 to 0876 $u$m and is converted to broadband via a scaling factor of 0.63 (Bair et al., 2019). MODSCAG/MODDRFS data covers most regions of

the TP, while SPIReS only covers the Indus river basin (Figure 1). Both products are aggregated to 0.125° spatial resolution and multi-year averaged seasonal scales.

### 3.3 Evaluation and analysis

The control simulation (i.e., case Sph_BCInt_DExt_PP in Table 1 is compared to MODIS data. The MODIS data represents

the snow states at the overpass time of about 10:30 am (local solar time). Thus, ELM simulated snow-related outputs are extracted from 10:00 am to 11:00 am (local solar time) to match the satellite observations and then aggregated to multi-year averaged seasonal scales: winter (DJF), spring (MAM), summer (JJA) and autumn (SON). Considering that the snow cover in summer and autumn is relatively low, this study focuses on winter and spring.

The impacts of snow grain shape and mixing state of LAP-snow on energy and water cycles in winter and spring are investigated from the various model configurations as listed in Table 1. Specifically, the impacts on the snow-related processes, energy budget, and water cycle are analyzed. The contributions of different influencing factors including snow grain shape, mixing state of LAP-snow, and sub-grid topographic effects are separated and compared based on the differences between various model configurations (see Table 2). Only the model grids with $f_{sno}$ larger than 0.01 in the control simulation are used

in the analysis. Note that RFs over all model grids are averaged (weighted by areas) to calculate the area-weighted average RFs over the TP (i.e., zero is involved in the calculation for any grids when snow is not present). The maximum absolute difference ($AD_{max}$), maximum absolute relative difference ($RD_{max}$), mean absolute difference ($AD_{mean}$), and mean absolute relative difference ($RD_{mean}$) are used to evaluate the effects of snow grain shape and mixing state. One-way analysis of variance (ANOVA) is used to determine whether the effects of snow grain shape and mixing state are statistically significant (if the

levels of statistical significance ($p$ values) are less than 0.05) or not.






**Table 2. Separating the contributions of different influencing factors based on the model experiments in Table 1.**

| Influencing Factor | Abbreviation | Snow grain shape | Difference between cases |
|---|---|---|---|
| Non-spherical shape (compared to spherical shape) | Non-spherical effect | Spheroid | Sphd_BCInt_DExt_PP – Sph_BCInt_DExt_PP |
| | | Hexagonal Plate | Hex_BCInt_DExt_PP – Sph_BCInt_DExt_PP |
| | | Koch Snowflake | Koc_BCInt_DExt_PP – Sph_BCInt_DExt_PP |
| Mixing state of BC-snow (Internal-External) | BC mixing state effect | Sphere | Sph_BCInt_DExt_PP – Sph_BCExt_DExt_PP |
| | | Koch Snowflake | Koc_BCInt_DExt_PP – Koc_BCExt_DExt_PP |
| Mixing state of dust-snow (Internal-External) | Dust mixing state effect | Sphere | Sph_BCExt_DInt_PP – Sph_BCExt_DExt_PP |
| | | Koch Snowflake | Koc_BCExt_DInt_PP – Koc_BCExt_DExt_PP |
| Sub-grid topographic effects | TOP effect | Sphere | Sph_BCExt_DInt_TOP – Sph_BCExt_DInt_PP |
| | | Koch Snowflake | Koc_BCExt_DInt_TOP – Koc_BCExt_DInt_PP |
| Combined effect | Combined effect | Koch Snowflake | Koc_BCExt_DInt_TOP – Sph_BCInt_DExt_PP |


## 4. Results

### 4.1 Comparison with remote sensing snow data

The ELM-simulated $f_{sno}$ in the control simulation shows good agreements with two MODIS products in winter (Figures 2, 3, S2 and S3). ELM and MODSCAG/MODDRFS have similar spatial patterns and show higher $f_{sno}$ in the western regions of the

TP in winter. The area-weighted average $f_{sno}$ for winter in ELM is within the range of the uncertainty bounds in MODSCAG/MODDRFS and SPIReS (Figure S3a) in winter. In spring, ELM underestimates $f_{sno}$ compared to MODSCAG/MODDRFS and SPIReS, although their spatial patterns are similar. Compared to the MODIS datasets, ELM simulation shows more smoothed spatial variability, which may be caused by the relatively coarse spatial resolution of the GSWP3 atmospheric forcing data and aerosol deposition data. Overall, the ELM simulation captures the spatial pattern of $f_{sno}$.

When the elevation is lower than 4.5 km, $f_{sno}$ values in both MODIS products and ELM increase with elevation (Figure 3a). $f_{sno}$ values at the three elevation intervals of 3.5-4.5 km, 4.5-5.5 km and >5.5 km have similar distributions for all three data (Figure 3a). At different elevation intervals, ELM shows similar mean values and ranges of $f_{sno}$ with MODSCAG/MODDRFS and SPIReS, especially in winter (Figure 3a and S2a).

ELM generally reproduces the overall spatial distribution of LAP-induced SAR, compared to MODSCAG/MODDRFS (Figure 4). In winter, ELM has a similar magnitude of SAR with MODSCAG/MODDRFS, while SPIReS has lower values. The area-





weighted average SAR for winter in ELM is within the range of the uncertainty bounds in MODSCAG/MODDRFS and SPIReS
(Figure S3b). In spring, ELM shows a similar change along the elevation gradient as compared to the MODSCAG/MODDRFS
(Figures 3d and S2d), but the estimated SAR are twice as large in ELM than in MODSCAG/MODDRFS and an order of

magnitude greater than SPIReS. The difference may be due to the overestimation of snow grain size and associated
underestimation of $\alpha_{sno}$ in ELM (Sarangi et al., 2020). Even the two MODIS products show large differences, probably due to
different assumptions used in their retrieval algorithm, limitations of multi-spectral sensor, and/or persistent cloud cover over
the TP. The coarse resolution and different time periods of the aerosol deposition data used in the ELM simulations can
contribute to the inconsistencies among the three datasets. Overall, ELM reasonably represents the snow cover distribution as

well as the LAP-induced SAR.

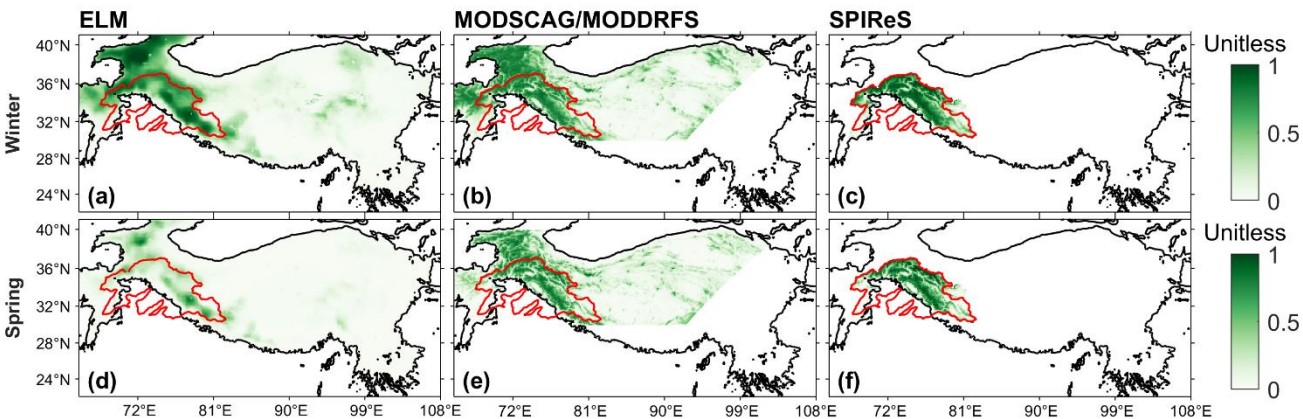

**Figure 2: Spatial distributions of $f_{sno}$ estimated from: (a,d) ELM, (b,e) MODSCAG/MODDRFS and (c,f) SPIReS for winter (a-c)**
**and spring (d-f).**




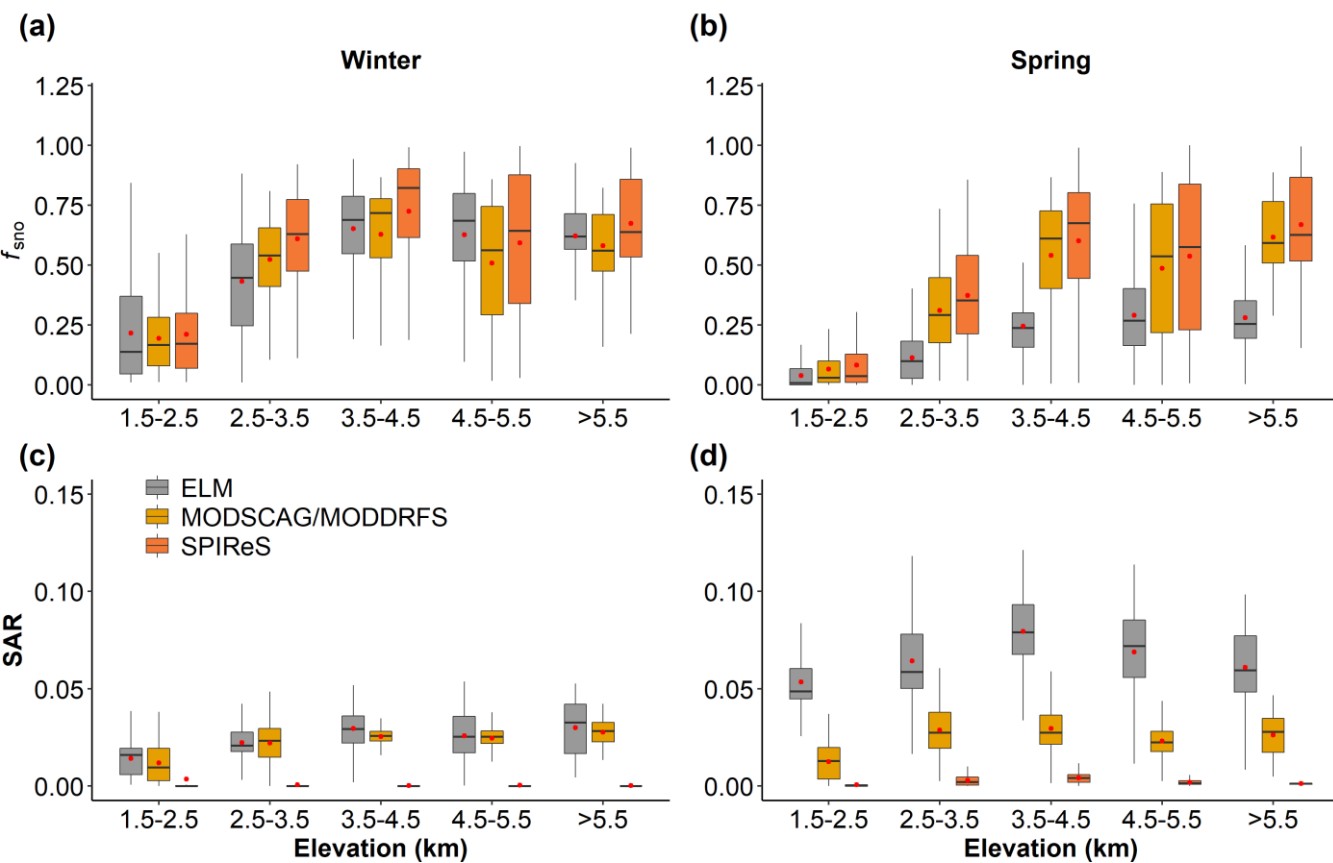

**Figure 3: Statistical distributions of (a,b) $f_{sno}$ and (c,d) SAR induced by LAPs for winter and spring in the overlapping areas (enclosed by the red lines in Figure 2) of ELM, MODSCAG/MODDRFS and SPIReS. The corresponding statistical distributions in the overlapping areas of ELM and MODSCAG/MODDRFS are shown in Figure S2.**


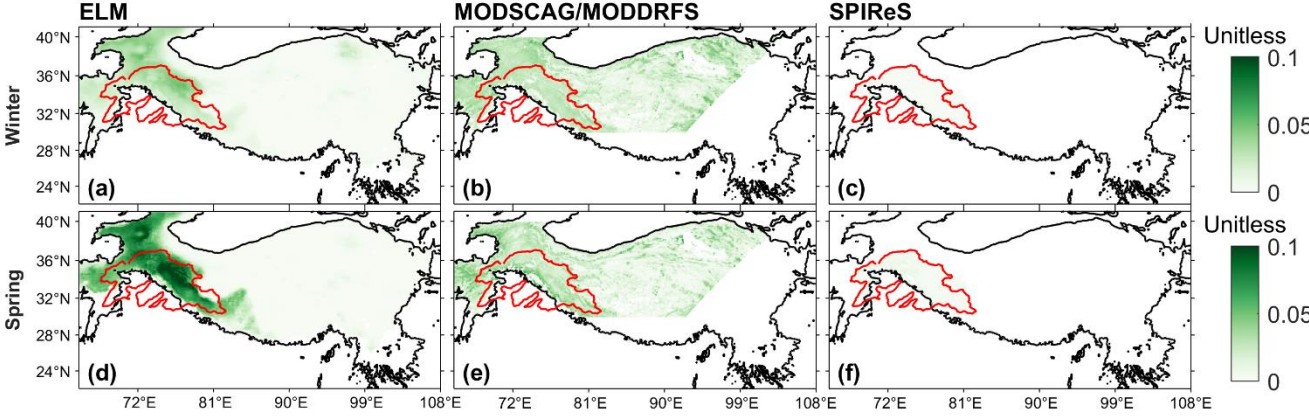

**Figure 4: Same as Figure 3, except for SAR.**



## 4.2 Radiative forcing induced by LAPs in snow

The RFs induced by different LAPs in the control simulation show divergent spatial distributions with seasonal variations (Figure 5). Western regions show larger RFs induced by all LAPs than the eastern regions (Figure 5a,c), while BC-induced RFs are larger in the southwestern regions (Figure 5b,e) and dust-induced RFs are larger in the northwestern regions (Figures 5c,f). Overall, spring has larger RFs than winter, because LAP-induced SAR is larger in spring caused by accumulated LAP concentration and resurfacing as snow melts (Figures 2-4) and the larger incident solar radiation over TP in spring than in

winter. In winter, BC overall has larger RFs than dust (Figures 5b,c), while dust has larger RFs in spring over the northwestern regions (Figures 5e,f). Table S1 summarizes the statistical maximum and mean values of RFs induced by different LAPs.

The LAP-induced RF values estimated in the ELM control case are within the range of the RF values reported by other studies. Table 1 summarizes the LAP-induced RF values for spring or non-monsoon season over the TP reported by previous studies

based on different climate models and different snow radiative transfer models. In spring, RFs by all LAPs in the control simulation are 0~21.9 W/m$^2$, which is close to the results in Qian et al. (2011) that showed RFs within 5–25 W/m$^2$ during spring. RFs by BC in the control simulation are about 0~9.2 W/m$^2$, which is identical to Ji (2011). Gertler et al. (2016) also reviewed different studies in the Himalaya and showed that BC-induced RFs range from 0 to 28.0 W/m$^2$. RFs by dust in the control simulation are about 0~10.2 W/m$^2$, which are similar to the results in Xie et al. (2018a). The area-weighted average

RFs by all LAPs, BC and dust are 1.3, 0.5 and 0.6 W/m$^2$, respectively in spring. Our results are also consistent with (Zhang et al., 2015), who found that snow LAPs induced RF has a seasonal peak in spring and a spatial maximum over the northwest TP with an average RF of 5 and 3.5 W/m$^2$, respectively, for BC and dust.

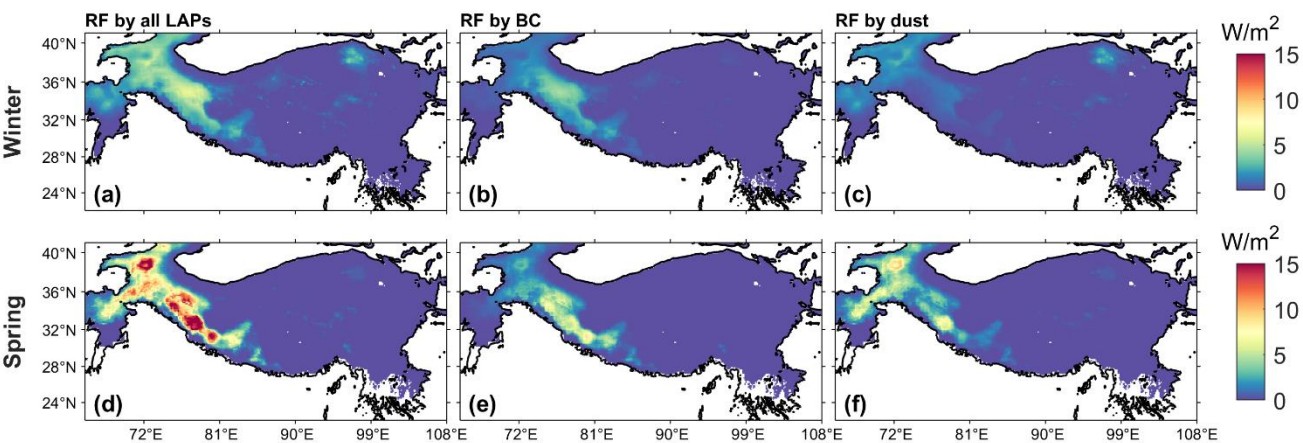

**Figure 5. Spatial distributions of RF values from (a,d) all LAPs (BC+dust), (b,e) BC and (c,f) dust for (a-c) winter and (d-f) spring.**





**Table 3. A summary of RF values of LAPs in snow at seasonal scales over the TP reported by the existing studies.**

| Model Name | Snow radiative transfer model | Aerosol type | RF value (W/m²) | Period | Reference |
|---|---|---|---|---|---|
| Community Atmosphere Model, version 3 (CAM 3) | SNICAR | BC | >20 in some parts of TP | Spring | (Flanner et al., 2007) |
| CAM 3.1 | SNICAR | BC+Dust | 0-9.5 or even >9.5 | Spring | (Flanner et al., 2009) |
| CAM 3.1 | SNICAR | BC+Dust | 5-25 | Spring | (Qian et al., 2011) |
| GEOS-Chem | Four-stream radiative transfer model | BC | Approximately 8-16 | Spring | (Kopacz et al., 2011) |
| GEOS-Chem | Geometric-optics surface-wave (GOS) approach | BC | 5-10 | Spring | (He et al., 2014) |
| CAM 5 | SNICAR | BC | 5 over the northwest TP | Spring | (Zhang et al., 2015) |
| | | Dust | 3 over the northwest TP | Spring | |
| Regional Climate Model version 4.3.4 | SNICAR | BC | 0-6 | November-April | (Ji, 2016) |
| CAM 4 | Bulk Aerosol Model parameterizations of the dust size distribution (BAM) | Dust | 0-20 or even >20 | Spring | (Xie et al., 2018a) |
| ELM v2.0 | Improved SNICAR-AD | BC+Dust | 0-21.9 | Spring | This study |

## 4.3 Impacts of snow grain shape and mixing state of LAP-snow

### 4.3.1 Impacts on snow related processes

Compared to the spherical shape, all three non-spherical grain shapes show larger $\alpha_{sno}$ and higher $f_{sno}$ in spring (Figure 6) and their differences are significant (ANOVA: $p < 0.05$). Among them, spheroid shape has the smallest differences from spherical shape and Koch snowflake has the largest differences from spherical shape (Figure 6). For instance, when $f_{sno} \geq 0.5$, $AD_{mean}$ in $\alpha_{sno}$ and $f_{sno}$ between Koch snowflake and spherical shape are 0.08 and 0.09, respectively (Table 4). The simulated snow water equivalent (SWE) also changes with snow grain shape and $AD_{mean}$ in SWE between Koch snowflake and spherical shape is 56.65 mm (Table 4). Similar results are obtained in winter (Figure S4), but they generally show smaller differences than those in spring. Similar results are obtained for spring and winter, thus only the results in spring are reported in the subsequent sections and the corresponding results for winter are reported in the supplementary material.

Compared to the external mixing of LAP-snow, internal mixing leads to smaller $\alpha_{sno}$ and lower $f_{sno}$ (Figure 7), although their differences are not significant (ANOVA: p > 0.1). For instance, when $f_{sno} \geq 0.5$, under spherical shape, $AD_{mean}$ in $\alpha_{sno}$ between internal and external mixing of BC-snow and dust-snow is approximately 0.009, while under Koch snowflake shape, $AD_{mean}$ caused by mixing state of BC-snow and dust-snow are 0.007 and 0.005, respectively. Similar results can be obtained for $f_{sno}$ from Figure 7e-h.




Both SAR and RFs induced by LAPs are sensitive to snow grain shape and mixing state of LAP-snow (Figure 8). The effects of snow grain shape and mixing state are more significant for regions with higher $f_{sno}$. Non-spherical shapes have smaller SAR and RFs induced by different LAPs, compared to the spherical shape. For instance, when $f_{sno} \geq 0.5$, all LAP-induced mean SAR and RF under spherical shape (Sph_BCInt_DExt_PP case) are 0.065 and 14.6 W/m$^2$, while those under Koch snowflake

(Koc_BCInt_DExt_PP case) are 0.051 and 12.1 W/m$^2$. Compared to external mixing, internal mixing of LAP-snow leads to larger SAR and higher RF values especially under spherical shape. For instance, when $f_{sno} \geq 0.5$, the dust-induced SAR and RF in the Sph_BCExt_DExt_PP case have mean values of 0.034 and 7.5 W/m$^2$, while the corresponding SAR and RF are larger in the Sph_BCExt_Dint_PP case with the mean values of 0.041 and 9.1 W/m$^2$, respectively. The effects of internal mixing become smaller for non-spherical grain shapes. For instance, the Koc_BCExt_DInt_TOP case has smaller SAR and RFs

compared to the Sph_BCExt_DInt_PP case.

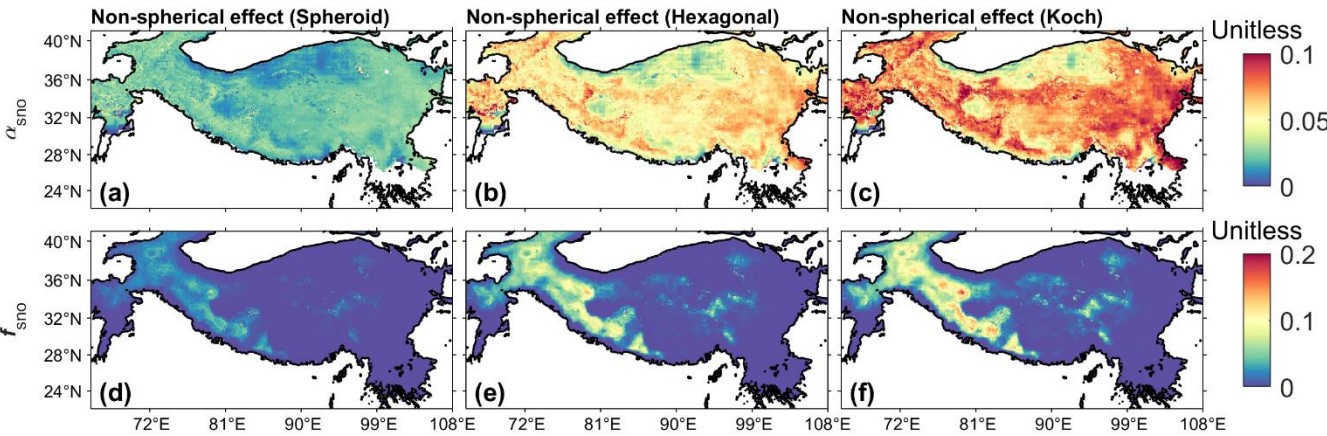

**Figure 6: The differences in $\alpha_{sno}$ (a-c) and $f_{sno}$ (d-f) between different snow grain shapes: (a,d) spheroid - sphere, (b,e) hexagonal plate - sphere and (c,f) Koch snowflake - sphere, for spring. The specific calculation methods are listed in Table 2.**






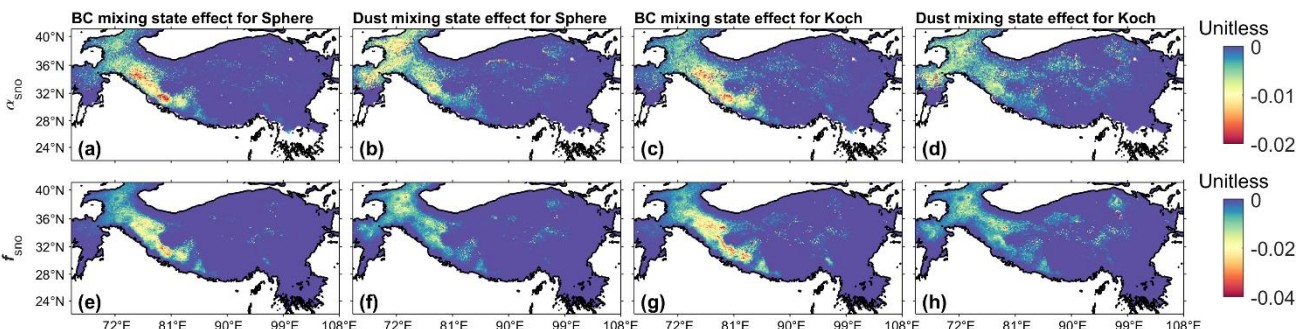

**Figure 7: The differences in $\alpha_{sno}$ (a-d) and $f_{sno}$ (e-h) between different mixing states (internal - external) of snow-LAP: (a,e) BC (sphere), (b,f) dust (sphere), (c,g) BC (Koch snowflake) and (d,h) dust (Koch snowflake) for spring. The specific calculation methods are listed in Table 2. The corresponding results for winter are shown in Figure S5.**



**Figure 8: Boxplots of SAR (a-b) and RF (c-d) from all LAPs, BC, and dust for different $f_{sno}$ values: (a,c) <0.5 and (b,d) ≥0.5 in spring under different cases listed in Table 1. For the case ID labelled in x-axis, the '_PP' suffix is omitted to keep them simplified. Red**
**circles represent the mean values. The corresponding results for winter are shown in Figure S6.**





### 4.3.2 Impacts on energy and water cycles

The impacts of snow grain shape and mixing state of LAP-snow on surface energy balance are large especially when $f_{sno}$ is high (Figures S7-S10). Generally as $f_{sno}$ increases, non-spherical grain shape has larger effects on surface energy balance terms

due to a larger change in $\alpha_{sno}$ and thus land surface albedo ($\alpha_{sur}$) (Figures S7-S8). Overall, Koch snowflake shape has the largest differences from spherical shape in all surface energy terms (Figures S7-S8). Due to the largest change of $\alpha_{sur}$, net solar radiation ($R_{net}^s$) of Koch snowflake has the largest difference from that of sphere (Table 4). For instance, $AD_{mean}$ and $RD_{mean}$ of $R_{net}^s$ are 17.6 W/m$^2$ and 0.17, respectively when $f_{sno} \geq 0.5$. The change of $R_{net}^s$ further leads to the change of surface temperature ($T_{sur}$), latent heat ($F_{lat}$) and sensible heat ($F_{sen}$) fluxes. Overall, the non-spherical grain shape has larger effects on

$F_{lat}$ than $F_{sen}$. Similarly, the effect of the mixing state of LAP-snow on surface energy balance overall increases with $f_{sno}$ (Figures S9-S10). The differences in the energy balance terms between the internal and external mixing of LAP-snow are generally larger for spherical shape than those for Koch snowflake shape. For instance, when $f_{sno} \geq 0.5$, under spherical shape, $AD_{mean}$ between internal and external mixing of BC-snow in $R_{net}^s$ is 2.3 W/m$^2$, while under Koch snowflake shape, the value is 2.1 W/m$^2$ (Table 4). Overall, the effects of mixing state of dust-snow are smaller than the effects of mixing state of BC-

snow. This effect has an opposite sign with the non-spherical shape effects and its magnitude is smaller than the effects of non-spherical grain shape (Figures S7-S10).

Water cycle is also affected by snow grain shape and mixing state of LAP-snow especially over the regions with high snow cover, due to snowmelt and altered $F_{lat}$ and $F_{sen}$ (Table 4 and Figures S11-S14). Non-spherical grain shape shows reduced

snowmelt, evapotranspiration (ET), and runoff, and Koch snowflake shape has the largest difference from spherical shape (Figures S11-S12). For example, when $f_{sno} \geq 0.5$, $AD_{mean}$ in snowmelt, ET and runoff between Koch snowflake and spherical shape are 0.43, 0.25, and 0.47 mm/day, respectively. The mixing state (i.e., the difference between internal and external mixing) of LAP-snow has smaller but opposite effects on water cycle than the effects from non-spherical shape, and snow grain shape also affects the magnitude of mixing state effects (Figures S13-S14). For instance, when $f_{sno} \geq 0.5$, $AD_{mean}$ between

internal and external mixing of BC-snow in snowmelt is 0.04 mm/day, while it is 0.07 mm/day under Koch snowflake shape.





**Table 4.** Statistics of the impacts of snow grain shape and mixing state of LAP-snow on snow-related processes, surface energy and water cycles in spring for the grid cells of $f_{sno} \geq 0.5$ in the control simulation.

| Category | Variable | Non-spherical effect (Koch) | | | | BC mixing state effect for sphere | | | | BC mixing state effect for Koch | | | | Dust mixing state effect for Sphere | | | | Dust mixing state effect for Koch | | | |
|---|---|---|---|---|---|---|---|---|---|---|---|---|---|---|---|---|---|---|---|---|---|
| | | $AD_{max}$ | $RD_{max}$ | $AD_{mean}$ | $RD_{mean}$ | $AD_{max}$ | $RD_{max}$ | $AD_{mean}$ | $RD_{mean}$ | $AD_{max}$ | $RD_{max}$ | $AD_{mean}$ | $RD_{mean}$ | $AD_{max}$ | $RD_{max}$ | $AD_{mean}$ | $RD_{mean}$ | $AD_{max}$ | $RD_{max}$ | $AD_{mean}$ | $RD_{mean}$ |
| Snow-related processes | $\alpha_{sno}$ | 0.096 | 0.15 | 0.076 | 0.11 | 0.023 | 0.03 | 0.009 | 0.01 | 0.018 | 0.03 | 0.007 | 0.01 | 0.016 | 0.02 | 0.009 | 0.01 | 0.010 | 0.02 | 0.005 | 0.01 |
| | $f_{sno}$ | 0.16 | 0.32 | 0.09 | 0.15 | 0.03 | 0.07 | 0.01 | 0.02 | 0.04 | 0.07 | 0.01 | 0.02 | 0.02 | 0.04 | 0.01 | 0.02 | 0.01 | 0.03 | 0.01 | 0.01 |
| | SWE (mm) | 101.65 | 0.54 | 56.65 | 0.26 | 18.33 | 0.11 | 7.04 | 0.03 | 17.48 | 0.13 | 6.45 | 0.03 | 10.20 | 0.05 | 4.83 | 0.02 | 8.66 | 0.05 | 3.81 | 0.02 |
| | SAR | 0.027 | 0.36 | 0.014 | 0.22 | 0.014 | 0.24 | 0.007 | 0.11 | 0.013 | 0.23 | 0.006 | 0.10 | 0.013 | 0.17 | 0.007 | 0.11 | 0.008 | 0.14 | 0.005 | 0.07 |
| | RF (W/m²) | 5.82 | 0.36 | 2.82 | 0.19 | 3.23 | 0.21 | 1.51 | 0.10 | 3.02 | 0.23 | 1.43 | 0.10 | 2.59 | 0.17 | 1.51 | 0.10 | 1.85 | 0.13 | 1.07 | 0.07 |
| Energy cycle | $\alpha_{sur}$ | 0.060 | 0.08 | 0.043 | 0.06 | 0.012 | 0.02 | 0.005 | 0.01 | 0.014 | 0.02 | 0.005 | 0.01 | 0.007 | 0.010 | 0.004 | 0.01 | 0.006 | 0.01 | 0.003 | 0.01 |
| | $R^S_{net}$ (W/m²) | 28.60 | 0.28 | 17.63 | 0.17 | 5.89 | 0.05 | 2.28 | 0.02 | 7.32 | 0.05 | 2.10 | 0.02 | 3.41 | 0.04 | 1.62 | 0.02 | 2.60 | 0.03 | 1.30 | 0.01 |
| | $T_{sur}$ (K) | 1.92 | - | 1.01 | - | 0.42 | - | 0.13 | - | 0.43 | - | 0.12 | - | 0.17 | - | 0.09 | - | 0.16 | - | 0.07 | - |
| | $F_{lat}$ (W/m²) | 13.41 | 0.74 | 7.16 | 0.27 | 2.79 | 0.15 | 0.97 | 0.04 | 2.75 | 0.08 | 0.86 | 0.03 | 1.71 | 0.12 | 0.67 | 0.03 | 1.37 | 0.06 | 0.50 | 0.02 |
| | $F_{sen}$ (W/m²) | 12.50 | 0.98 | 4.93 | 0.56 | 2.82 | 0.23 | 0.71 | 0.08 | 2.23 | 0.16 | 0.54 | 0.06 | 1.11 | 0.17 | 0.49 | 0.06 | 0.86 | 0.09 | 0.32 | 0.04 |
| Water cycle | Snowmelt (mm/day) | 2.07 | 0.42 | 0.43 | 0.08 | 0.21 | 0.04 | 0.04 | 0.01 | 0.41 | 0.05 | 0.07 | 0.01 | 0.27 | 0.06 | 0.04 | 0.01 | 0.39 | 0.05 | 0.06 | 0.01 |
| | ET (mm/day) | 0.46 | 0.74 | 0.25 | 0.27 | 0.10 | 0.15 | 0.03 | 0.04 | 0.10 | 0.08 | 0.03 | 0.03 | 0.06 | 0.12 | 0.02 | 0.03 | 0.05 | 0.06 | 0.02 | 0.02 |
| | Runoff (mm/day) | 2.75 | 0.51 | 0.47 | 0.16 | 0.83 | 0.14 | 0.06 | 0.02 | 0.96 | 0.23 | 0.07 | 0.02 | 0.63 | 0.16 | 0.05 | 0.02 | 0.37 | 0.11 | 0.05 | 0.02 |





**4.4 Combined effects of snow grain shape, mixing state of LAP-snow and sub-grid topography**

The snow grain shape, mixing state of LAP-snow, and sub-grid topography can all affect $\alpha_{sur}$ and thus the surface energy balance and water cycle. Taking $R_{net}^s$ as an example, their combined effects on $R_{net}^s$ may be negative or positive and vary from
-29.7 to 12.2 W/m$^2$, depending on the sign and magnitude of the effect of individual influencing factors (Figure 9). The non-spherical shape effects over TP are negative (Figure 9b), while the effects of mixing state of LAPs (BC and dust) are positive for both sphere and Koch snowflake with a smaller magnitude (Figure 9e-h). Different from the snow grain shape and mixing state, topography can affect the surface radiation budget over both the snow-covered and snow-free complex terrain (Figure 9c,d) due to the shadowing effects and multi-scattering from adjacent terrain. The TOP effects can be positive or negative,
depending on the local topographic features. Thus interactions among the three influencing factors are complex and nonlinear, and their effects can be supplementary or cancelling. For instance, the TOP effects can be different under different snow grain shapes (Figure 9c,d) and they vary from -5.2 to 14.1 W/m$^2$ under spherical shape and from -7.5 to 16.9 W/m$^2$ under Koch snowflake shape (Figure 9c-d).






Figure 9: Spatial distributions of the change of net solar radiation ($R_{net}^s$) contributed by individual influencing factors and their combined effects in spring, which are derived based on Table 2. The corresponding results in winter are shown in Figure S15.

## 5. Discussion

The magnitude and spatial distribution of snow cover in the ELM simulations are similar to the MODIS estimates in winter, but are underestimated in spring. This underestimation is also found in CLM4.5 as reported by (Xie et al., 2018b), which used the same snow cover parameterizations of (Swenson and Lawrence, 2012) as in ELM v2.0. The underestimation may be caused





by the use of a constant snow accumulation ratio, empirical snowmelt shape factor, and the complex vegetation-snow
interaction processes of snow interception and dynamical removal from canopy (Xie et al., 2018b). The canopy-gap effects on
$f_{sno}$ are not accounted for in ELM v2.0. Besides, uncertainties of the atmospheric forcing data can affect the accuracy of the
ELM snow simulations (Wang et al., 2020b). Although ELM v2.0 is run at a 0.125° spatial resolution, 0.5° GSWP3v1
atmospheric data is used in our simulations, which leads to a smoothed spatial variability of snow simulated by ELM v2.0
compared to MODIS data (Figures 2 and 4). The wind-blowing of snow and the subsequent sublimation process also frequently
affect the snow dynamics and are unaccounted for in ELM v2.0 (Orsolini et al., 2019; Xie et al., 2019).

There are some differences in SAR between ELM v2.0 and MODIS data. ELM v2.0 shows a similar magnitude to
MODSCAG/MODDRFS in winter, but is larger than MODSCAG/MODDRFS in spring, while SPIRES shows lower values
than ELM v2.0 in both winter and spring. The ELM v2.0 results are in agreement with He et al. (2018a), showing that BC-
induced SAR can be above 0.1 in the non-monsoon season, and comparable to the field measurements-based analysis in the
central and western Himalayas of (Gul et al., 2021). The use of prescribed climatological aerosol deposition data may in part
contribute to the inconsistencies between model results and remote sensing data. The simulation biases of snow accumulation,
melting, refreezing, compaction, aging, and water transport across snow layers in ELM v2.0 can affect the SAR estimates.
Note that there are large differences in SAR between MODSCAG/MODDRFS and SPIRES (Bair et al., 2021a), because snow
estimates from MODIS are largely affected by frequent cloud cover over TP, off-nadir effects and reflectance measurement
errors (Bair et al., 2021c). Although $f_{sno}$ estimated by MODSCAG/MODDRFS and SPIReS is relatively reliable, there are still
large uncertainties in the estimated snow grain size and LAP content and thus SAR due to the limited capability of multi-
spectral sensors to distinguish between darkening from micro-scale topography and LAPs in low concentrations (Bair et al.,
2021b). The ongoing and upcoming hyperspectral satellites such as PRISMA (PRecursore IperSpettrale della Missione
Applicativa) mission, the Surface Biology and Geology (SBG) led by NASA and the Copernicus Hyperspectral Imaging
Mission for the Environment (CHIME) led by European Space Agency (ESA) will be promising for improving the remotely
sensed estimates of LAP-induced SAR. Our study focuses on the relative sensitivity rather than absolute accuracy, and thus
the abovementioned uncertainties are expected to have small influences on our results.

The heterogeneous spatial distribution of RFs estimated by ELM v2.0 is overall comparable to the reported values in the
existing study. Spring has larger RF values from different LAPs than winter (Figure 5; Qian et al., 2011), due to larger LAP
concentrations accumulated during snowmelt process (Kang et al., 2020) and larger incident solar radiation. Dust has a larger
RF value than BC in spring over the northwestern regions of the TP (Figure 5f), as reported in (Sarangi et al., 2020), and is
related to the regional difference of the source, transport, and deposition of BC and dust. However, different studies used
different regional or global land surface models that were coupled or uncoupled with atmosphere models to simulate the SARs
and estimated RFs for different time periods at different spatial resolutions. These models represent the snow processes with
different complexities, account for different LAP types, and have different assumptions about LAP effects on $\alpha_{sno}$ (Gertler et





al., 2016). The simulations were run for different time periods, and thus it is difficult to resolve the issues of spatio-temporal mismatch between model simulations and *in situ* measurements (Qian et al., 2015). These studies also reported the RF values

at different temporal scales from instantaneous, at the overpass time of remote sensing observations, daily, monthly, seasonal, or annual scales.

Snow grain shape and mixing state of LAP-snow play an important role in snow processes, surface energy balance and water cycle. Snow grain shape has a large effect on $\alpha_{sno}$ and the Koch snowflake shows the largest differences from sphere, as

reported in (He et al., 2018a). Mixing state of LAP-snow also has large impacts on SAR induced by BC and dust (He et al., 2018a; He et al., 2019). With the treatment of Koch snowflake shape, internal mixing of LAP-snow has smaller effects on $\alpha_{sno}$ than that of the spherical shape (Figure 6; He et al., 2018a). Their impacts on $\alpha_{sno}$ further affect the surface energy balance and water cycles (Section 4.3.2). For example, the Koch-snowflake shape decreases $R_{net}^s$ by up to 30 W/m$^2$ compared to the spherical shape (Table 4). Long-term field measurements of snow grain characteristics, $\alpha_{sno}$ and LAP concentrations over the

TP are needed to evaluate the model simulations and advance our understanding of snow grain shape and mixing state effects. Topography also changes the apparent $A_{sur}$ and solar radiation absorbed by the surface, and in turn affects snow dynamics (Hao et al., 2021a; Hao et al., 2021b; Hao et al., 2018). Compared to the negative effects of non-spherical shape on $R_{net}^s$, topography generally has positive effects on $R_{net}^s$ with a larger magnitude than mixing state of LAP-snow (Figure 9). Their effects can be additive or cancelling out each other and thus the combined effects may be cooling or warming (Figure 9), which depend on

the specific snow grain shape, mixing state, and local topographic features. A novel topography-based sub-grid structure (topounit) (Tesfa and Leung, 2017) has been added in ELM. Furthermore, jointly including the improved snow albedo model from this study, TOP solar radiation parameterization of Hao et al. (2021b) within the topounit structure, and considering the downscaling of snowfall based on topounit from (Tesfa et al., 2020) will be promising to improve the simulation of snow dynamics.


There remain some uncertainties in parameterizing the effects of LAPs on $\alpha_{sno}$ in ELM. A mono-disperse (uniform) snow grain size distribution is assumed in the new treatments of Section 2.2, although non-uniform (e.g., lognormal) snow size distributions were observed and may have significant impacts on $\alpha_{sno}$ (Saito et al., 2019). The typical properties of BC and dust, such as refractive index, size distribution, and particle morphology from observations were adopted in the

parameterizations of ELM (He et al., 2019; He et al., 2017). The maximum allowable BC and dust concentrations are 1000 ppb and 1000 ppm, respectively (He et al., 2019; He et al., 2017). The parameterizations assume that BC/dust particles randomly distribute internally in the snow grains, but a non-random distribution of BC/dust in snow grains may affect the magnitude of SAR (Dombrovsky and Kokhanovsky, 2020; Shi et al., 2021). The parameterization for the effects of internally-mixed BC (or dust) assumes no presence of dust (or BC) in the snow. In order to avoid the possible overestimations of SAR,

the simultaneous internal mixing of BC and dust in snow grains is not included in this study, which requires future investigations. It is also assumed that different types of LAPs are externally mixed with each other. However, different types





can also be internally mixed (i.e., coating or attachment) with each other (Chung et al., 2012; Pu et al., 2021), leading to an enhanced absorption through the so-called lensing effects (Bond et al., 2006; Lack et al., 2012). For instance, BC absorption enhancement is strongly relevant to the amount of coating material and is also source and region dependent (Liu et al., 2015).

The effects of BrC with high uncertainties are neglected in our study. In contrast to BC, sources, chemical composition and optical properties of BrC are still poorly understood (Wu et al., 2016) and its effects on RF and its contributions to SAR remain highly uncertain (Beres et al., 2020). With some assumptions of the photochemical and optical properties of BrC in SNICAR within the Community Land Model (version 4), Brown et al. (2021) found that the global (and TP) mean RF of BrC in snow can be comparable to that of BC. A more realistic representation of the impacts of snow grain properties and diverse LAPs on

$\alpha_{sno}$ that is validated against field measurements and remote sensing data in land surface models (e.g., ELM) is required for climate change studies.

There are some limitations of this study. Our sensitivity tests are based on assumptions of a spatiotemporally homogenous snow grain shape and mixing state of LAP-snow and currently there is no knowledge about the specific snow grain shape and

mixing state of dust-snow regionally or globally. Only four typical types of snow grain shapes are included in the analysis, but the real snow grain shape may be more complicated and irregular (Kokhanovsky et al., 2005). Besides, in reality, both the snow grain shape and mixing state of LAP-snow are spatially-inhomogeneous and time-varying (Räisänen et al., 2015). For instance, dust may be partially internally and externally mixed with snow grains and simply assuming external mixing of dust-snow could underestimate the dust effects, while assuming fully dust-snow internal mixing will overestimate the dust effects

(Shi et al., 2021). The land-atmosphere interaction is neglected by performing offline ELM simulations. The impacts of BrC and snow algae are also excluded in this study, which may be comparable to BC and dust effects (Brown et al., 2021; Dang and Hegg, 2014; Di Mauro, 2020; Ganey et al., 2017).

## 6. Conclusions

Snow albedo is sensitive to snow grain shape and mixing state of LAP-snow. This study implemented the computationally

efficient parameterizations for non-spherical grain shape (i.e., spheroid, hexagonal plate, and Koch snowflake) and the internal mixing of dust-snow in the snow radiative transfer model (i.e., SNICAR-AD) in ELM v2.0. The ELM control simulations show similar snow distribution as two MODIS snow products (MODSCAG/MODDRFS and SPIReS) with some noticeable statistical differences in spring. The LAP-induced RFs in the control simulations in spring are also within the range of the reported values in previous studies, though considerably higher than MODIS snow products in spring. All the snow-related

processes, surface energy balance and water cycles are sensitive to the treatment of snow grain shape. The Koch snowflake shape shows the largest difference from the default spherical shape treatment in ELM. The impacts of the mixing state of LAP-snow are smaller than the non-spherical shape effects, which also depend on snow grain shape. The effects of non-spherical shape, mixing state of LAP-snow, and sub-grid topography on snow and surface fluxes have different signs and magnitudes.



Their combined effects are complex, non-linear and may be negative or positive depending on the specific snow grain shape,
mixing state, and local topographic features. Overall, the changes of net solar radiation in spring due to individual and
combined effects range from -28.6 to 16.9 $W/m^2$ and -29.7 to 12.2 $W/m^2$, respectively. This study advances our understanding
of uncertainties in snow albedo modeling and its effects on surface energy and water cycles, and offers guidance for improving
the simulations of snow processes and RF estimates in ESMs. Future efforts are needed to couple the impacts of BrC and snow
algae and investigate the climate effects of LAPs in snow via land-atmosphere coupling.


**Code and data availability**

The    description    and    codes    of    E3SM    v2.0    (including    ELM    v2.0)    are    publicly    available    at
https://doi.org/10.11578/E3SM/dc.20210927.1 and https://github.com/E3SM-Project/E3SM/releases/tag/v2.0.0 (last access:
12 April 2022), respectively. Starting from ELM 2.0, the model codes for snow albedo modeling improvements described in
this paper is available at https://doi.org/10.5281/zenodo.6324131, and code to reproduce all results and plot all figures is
publicly        available        at        https://github.com/daleihao/Snow_Albedo_Parameterization_in_ELM        and
https://doi.org/10.5281/zenodo.6321316.

**Author contributions**

DH designed the study, implemented the parameterization, performed the simulations, analyzed the results, and drafted the
original manuscript. GB designed the study, discussed the results, and edited the manuscript. CH and CD helped with the snow
albedo parameterization and edited the manuscript. EB and KR provided the remote sensing data and edited the manuscript.
HH, YG, HW, YQ, and LRL discussed the results and edited the manuscript. All authors contributed to improving the
manuscript.

**Competing interests**

The authors declare that they have no conflict of interest.

**Acknowledgements**

This research was conducted at Pacific Northwest National Laboratory, which is operated for the U.S. Department of Energy
by Battelle Memorial Institute under contract DEAC05-76RL01830. This research used resources of the National Energy
Research Scientific Computing Center (NERSC), a DOE Office of Science User Facility supported by the Office of Science
of the U.S. Department of Energy under contract no. DE-AC02-15 05CH11231. The reported research also used DOE's



Biological and Environmental Research Earth System Modeling program's Compy computing cluster located at Pacific Northwest National Laboratory. We thank Jeff Dozier for his help with snow albedo modeling. The MODSCAG/MODDRFS data is available at ftp://snowserver.colorado.edu/pub/fromRittger/products/Indus. The SPIReS data is available at
ftp://ftp.snow.ucsb.edu/pub/org/snow/products/SPIRES/Indus/.

## Financial support

This research has been supported by the U.S. Department of Energy, Office of Science, Office of Biological and Environmental Research, Earth System Model Development program area, as part of the Climate Process Team projects and the U.S. National
Oceanic and Atmospheric Administration (NOAA, grant no. NOAA-OAR-CPO-2019-2005530).

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
