# Peer review of "Improving snow albedo modeling in E3SM land model (version 2.0) and assessing its impacts on snow and surface fluxes over the Tibetan Plateau"

_Geoscientific Model Development, 2022_

## Author Comment (AC1)

**Reviewer 1:**

The mixing state of LAPs in snow has large impacts on LAP-induced snow albedo reduction and surface radiative forcing (RF). However, most land surface models assume that snow grain shape is spherical and LAPs are externally mixed with the snow grains. On this background, the authors improved the snow radiative transfer model in the land model (ELM v2.0) of the Energy Exascale Earth System Model version 2.0 (E3SM v2.0) by considering non-spherical snow grain shapes (i.e., spheroid, hexagonal plate and Koch snowflake) and internal mixing of dust-snow and systematically evaluates the impacts on surface energy and water balances over the Tibetan Plateau (TP). In general, this study is well written and can advance understanding of the role of snow grain shape and mixing state of LAP-snow in land surface processes and offer guidance for improving snow simulations and RF estimates in Earth system models under climate change. I think this manuscript can be accepted if address the following questions.

We appreciate your comments and suggestions, and we have revised the manuscript accordingly.

1. The authors spent much of this manuscript on comparing the modeling results with remote sensing snow data under the control simulation using the previous settings, which was not the main concern of this manuscript. This analysis only demonstrated the capability of the original land model in snow cover fraction or SAR simulation, but can not reflect the improvement of your work. So, I suggest the results based on the original land model and your improved version can be simultaneously compared with remote sensing data to highlight the improvement of your simulation.

As suggested, we have now added a comparison of the original (ELM_Control) and new versions (ELM_New) against remote sensing data. In this revised manuscript, we used the new versions of STC-MODSCAG/STC-MODDRFS and SPIReS data. The results show that ELM_New has smaller biases of snow cover fraction than ELM_Control in spring, compared to two remote sensing data (Figures R1). More specifically, ELM_New reduces the underestimations of snow cover fraction at different elevations and the area-weighted average values of snow cover fraction in ELM_Control (Figures R2 and R3). Compared to the mean value of the two MODIS datasets, ELM_New reduces 0.014 (13.6%) of the bias of

ELM_Control in the area-weighted average $f_{sno}$ for spring (Figure R2a). However, both ELM_Control and ELM_New show large differences in the spatial distribution of LAP-induced SAR, compared to the two remote sensing datasets, which are also quite different from each other (Figure R3c-d). We have added the corresponding results in Line 282-326 of the revised manuscript. We also updated the corresponding contents in the abstract, discussion and conclusions of the revised manuscript.

[Figure]

**Figure R1: (a,c) The $f_{sno}$ bias ($\delta_{ELM\_Control}$) of ELM_Control compared to the mean value of STC-MODSCAG and SPIReS, and (b,d) the difference ($|\delta_{ELM\_New}|-|\delta_{ELM\_Control}|$) between the absolute values of the biases of ELM_New ($\delta_{ELM\_New}$) and ELM_Control ($\delta_{ELM\_Control}$) for winter (a-b) and spring (c-d). The negative values (blue color) in (b,d) show that ELM_New has a smaller bias than ELM_Control. The areas with $f_{sno}$ smaller than 0.01 are masked. This figure corresponds to Figure 2 in the revised manuscript.**

[Figure]

**Figure R2:** The area-weighted average of (a) $f_{sno}$ and (b) SAR induced by LAPs for winter and spring from ELM, STC-MODSCAG/STC-MODDRFS, and SPIReS. The bar width represents the uncertainty bounds of STC-MODSCAG and SPIReS from Bair et al. (2021a). This figure corresponds to Figure 3 in the revised manuscript.

[Figure]

**Figure R3: Statistical distributions of (a,b) $f_{sno}$ and (c,d) SAR induced by LAPs under different elevation ranges for winter and spring from ELM, STC-MODSCAG/STC-MODDRFS, and SPIReS. This figure corresponds to Figure 4 in the revised manuscript.**

2. In the Abstract, the authors said "Compared with two remote sensing ……, the control ELM simulation with the default settings of spherical snow grain shape, internal mixing of BC-snow, external mixing of dust-snow and without TOP can capture the overall snow distribution reasonably." So my question is what is your contribution? From your abstract, I can not find better simulation results after improving these snow microphysics properties. I think the authors should highlight the advances in reproducing the satellite observations after the improvements.

As mentioned in our responses to Q1, we have added a comparison of ELM_Control and ELM_New against the MODIS data in the revised manuscript and found that with the new model features, the simulation of snow cover fraction is generally improved. However, large biases remain in the LAP-induced SAR and large differences are noted between two remote-sensing products.

Although model improvements are generally small or mixed, a main contribution of this study is to advance understanding of uncertainties in snow albedo modeling and their impacts on energy budget and water cycle by systematically evaluating the impacts of non-spherical snow grain shape and internal mixing of LAP-snow across the full range of $f_{sno}$ on surface energy balance and water cycles over the Tibetan Plateau (See Figures 10 and 11).

3. The two used remote sensing data of ASR have larger differences themselves, so I wonder whether it is appropriate to use these two data in comparing with your simulations. Maybe only using MODSCAG/MODDRFS data is better from your results.

The two remote sensing datasets (i.e., STC-MODSCAG/STC-MODDRFS and SPIReS) are based on different models/algorithms (Bair et al., 2021) and no study, to our knowledge, has evaluated the relative accuracy of these datasets due to limited field measurements in the Tibetan Plateau. A comprehensive evaluation of the two remote sensing products based on field measurements is needed to determine the accuracy of each product, which is beyond the scope of this study. Thus, we used both STC-MODSCAG/STC-MODDRFS and SPIReS to provide an observational bound for comparison with the SAR estimates of ELM. We also discussed the uncertainties of both remote sensing data and ELM simulations in Line 509-534 of the discussion part of the revised manuscript.

4. From the results, I found the snow grain shape and TOP have larger impacts on snow cover fraction or SAR simulation, while the impact of mixing sate is relatively lower. So my question is why you spent most of your content on snow grain shape and mixing sate, but the impact of TOP was only simply discussed in the last Section 4.4.

This work aims to improve the snow radiative transfer model in ELM v2.0 by accounting for non-spherical snow grain shape and internal mixing of LAP-snow and systematically evaluate their impacts on surface energy balance and water cycle over the Tibetan Plateau. Thus, we focused on the impacts on the non-spherical snow grain shape and mixing state of LAP-snow. Indeed, apart from snow grain shape and mixing state of LAP-snow, topography also changes the

apparent surface albedo and solar radiation absorbed by the surface, and in turn affects snow dynamics. Compared to ELM_Control, the individual contributions of non-spherical snow shape, mixing state of LAP-snow, and local topography to the change of snow and surface fluxes have different signs and magnitudes, and their combined effects may be negative or positive due to complex and non-linear interactions among the factors. Considering that we have already documented the TOP parameterization and its impacts on energy budget and snow processes in another published GMD paper (Hao et al., 2021), we mainly investigated the interactions between the improved snow radiative transfer model and sub-grid topographic effects (TOP) in this work to minimize repetition. We added the finding about the TOP effects in the abstract (Line 35-38), discussion (Line 464-474) and conclusions (Line 556-559) in the revised manuscript.

5. When discussing the impacts on water cycles, the authors only spent little content while this part was very important. In addition, this is no figure in the manuscript in this part. I think more discussions are needed because the impact of snow grain shape and mixing state of LAP-snow on SAR or RF have been fully discussed in previous studies while the impact on water cycles is little mentioned.

The revised manuscript includes one table to present the impacts on energy budget and water cycle. Considering that Figures R4 and R5 below show the important impacts of snow grain shape on surface energy balance and water cycle, we have transferred them from the supplementary materials of the original manuscript to the main text of the revised manuscript. We also added more analysis and discussion on the impacts on energy and water cycles under different snow cover fractions and more mechanistic explanations in Line 403-435 of the revised manuscript.

[Figure]

Figure R4: The boxplots of the differences (Δ) in surface energy budget terms: (a) surface albedo ($\alpha_{sur}$), (b) net solar radiation ($R_{net}^S$), (c) surface temperature ($T_{sur}$), (d) latent heat flux ($F_{lat}$) and (e) sensible heat flux ($F_{sen}$), between different snow grain shapes: spheroid – sphere, hexagonal plate – sphere and Koch snowflake – sphere under different snow cover fractions ($f_{sno}$) for spring. See Table 2 for the specific calculation methods. This figure corresponds to Figure 10 in the revised manuscript.

[Figure]

**Figure R5: Same as Figure S4, except for hydrological terms: (a) snowmelt, (b) ET, and (c) runoff. This figure corresponds to Figure 11 in the revised manuscript.**

6. In the Discussion, the authors spent much time on discussing why the snow cover was underestimated in spring. As mentioned in the Discussion, the underestimation may be caused by the use of a constant snow accumulation ratio, empirical snowmelt shape factor, and the complex vegetation-snow interaction processes of snow interception and dynamical removal from canopy. So I think this was not closely related to the contributions of this manuscript. While the discussions from Lines 468-517 are the main limitations in your work and the directions the future work moves on. I suggest the authors can discuss the limitations in the first place, then simply discuss the underestimation.

We followed this suggestion to revise the discussion Section 5 of the revised manuscript. We first discussed the main contributions of this work on the impacts of snow grain shapes and mixing states of LAP-snow, compared with the existing studies and stated the limitations and prospects for further improvements of snow albedo parameterizations. We further discussed the

uncertainties of ELM simulations compared to the remote sensing data and the existing issues of both ELM and remote sensing data.

**References**

Bair, E. H., J. Dozier, C. Stern, A. LeWinter, K. Rittger, A. Savagian, T. Stillinger, and R. E. Davis (2022), Divergence of apparent and intrinsic snow albedo over a season at a sub-alpine site with implications for remote sensing, The Cryosphere, 16(5), 1765-1778, doi: 10.5194/tc-16-1765-2022.

Hao, D., Bisht, G., Gu, Y., Lee, W. L., Liou, K. N., and Leung, L. R.: A parameterization of sub-grid topographical effects on solar radiation in the E3SM Land Model (version 1.0): implementation and evaluation over the Tibetan Plateau, Geosci. Model Dev., 14, 6273-6289, 2021.

---

## Author Comment (AC2)

**Reviewer 2:**

This study presents an overview of the impact of several grain shapes, the mixing state of light absorbing impurities and sub-grid topographic effects on the surface energy and water cycles in the Tibetan Plateau using ELM. The authors illustrate that non-spherical shapes and sub-grid topographic effects impact the surface energy and water balance considerably during spring, while the impact of the mixing state of light absorbing impurities is smaller. The manuscript is well written and the topic is relevant. The impact of the mixing state of light absorbing impurities and topographic effects, and to a lesser degree also grain shape, is often overlooked in ESMs and climate models. This manuscript thus provides a necessary overview of the importance of these processes for such models.

We appreciate your insightful comments and suggestions. We have revised the manuscript carefully.

The following comments should help with solving the remaining issues, with, for example, P1 L1 meaning page 1, line 1.

**General comments**

The comparison with satellite observations is limited to the control run only and could be expanded to some of the other experiments as well. It would be especially interesting to see how well ELM performs for the 'combined effect' experiment. Also, if the authors could make a figure showing the difference between model and observations could help visualizing the results.

As suggested, we have now added a comparison of the original (ELM_Control) and improved versions (ELM_New) against remote sensing data. In this revised manuscript, we used new versions of STC-MODSCAG/STC-MODDRFS and SPIReS data. The results show that ELM_New has smaller biases of snow cover fraction than ELM_Control in spring, compared to the two remote sensing datasets (Figures R1). ELM_New reduces the underestimations of snow cover fraction at different elevation gradients and the area-weighted average values of snow cover fraction in ELM_Control (Figures R2 and R3). Compared with the mean value of the two

MODIS data, ELM_New reduces 0.014 (13.6%) of the bias of ELM_Control in the area-weighted average $f_{sno}$ for spring (Figure R2a). However, both ELM_Control and ELM_New show large differences in the spatial distribution of LAP-induced SAR, compared to the two remote sensing datasets (Figure R3c-d). We have added the corresponding results in Line 282-326 of the revised manuscript. We also updated the corresponding contents in the abstract, discussion and conclusions of the revised manuscript.

[Figure]

**Figure R1: (a,c) The $f_{sno}$ bias ($\delta_{ELM\_Control}$) of ELM_Control compared to the mean value of STC-MODSCAG and SPIReS, and (b,d) the difference ($|\delta_{ELM\_New}|-|\delta_{ELM\_Control}|$) between the absolute values of the biases of ELM_New ($\delta_{ELM\_New}$) and ELM_Control ($\delta_{ELM\_Control}$) for winter (a-b) and spring (c-d). The negative values (blue color) in (b,d) show that ELM_New has the smaller bias than ELM_Control. The areas with $f_{sno}$ smaller than 0.01 are masked. This figure corresponds to the Figure 2 in the revised manuscript.**

[Figure]

**Figure R2: The area-weighted average of (a)** $f_{sno}$ **and (b) SAR induced by LAPs for winter and spring from ELM, STC-MODSCAG/STC-MODDRFS, and SPIReS. The bar width represents the uncertainty bounds of STC-MODSCAG and SPIReS from Bair et al. (2021a). This figure corresponds to the Figure 3 in the revised manuscript.**

[Figure]

**Figure R3: Statistical distributions of (a,b) $f_{sno}$ and (c,d) SAR induced by LAPs under different elevation ranges for winter and spring from ELM, STC-MODSCAG/STC-MODDRFS and SPIReS. This figure corresponds to Figure 4 in the revised manuscript.**

It is not clear to me what grain size is chosen in this study and why. Furthermore, it is also good to notice that a homogeneous snow grain size that does not vary in time, as is employed in ELM if I am not mistaken, could have a significant impact on the results. Wet and dry snow metamorphism, refreezing and the presence of slush or ice can all impact the grain size and albedo. This may explain for a large part the differences observed with remote sensing observations. Similarly, it is not clear to me what concentrations of LAPs are chosen.

In this study, the snow grain size is not a prescribed parameter. SNICAR-AD can mechanistically simulate the snow aging process, i.e., the evolution of the snow grain size, by considering the dry snow metamorphism, liquid water-induced metamorphism, refreezing of liquid water, and addition of freshly-fallen snow (Flanner et al., 2007). Besides, SNICAR-AD in ELM can also prognostically simulate the change of LAP concentrations in different layers (Flanner et al., 2007). After LAPs are deposited from the atmosphere, LAPs are redistributed

based on meltwater drainage through the snow column and snow layer combination and subdivision. Thus, both the LAP concentration and snow grain size in the ELM simulations vary with space and time. We clarified these points above in Line 133-134 of the revised manuscript.

This study investigates several grain shapes. It is not clear to me, however, why these grain shapes are chosen in particular. Are these grain shapes often occurring on the Tibetan Plateau? If the authors could show that these shapes are relevant, it would strengthen the results of this study. Also, what grain shape will you assume in the final ELM version?

The three non-spherical shapes chosen in this study are the typical snow grain shapes that have been previously used to capture the major morphological characteristics of observed snow grain structures in the real snowpack (Dominé et al., 2003; Erbe et al., 2003; Liou et al., 2014). Moreover, the simulated snow albedo with the assumptions of the three non-spherical snow shapes showed better agreement with the observations (He et al., 2018). However, note that there are very limited field observations of snow grain shapes over the Tibetan Plateau, which prevents an accurate prescription of snow shape in ELM. Therefore, we have conducted sensitivity analyses to quantify the impact and uncertainty caused by these three non-spherical snow grain shapes. Only four typical types of snow grain shapes are included in the analysis, but the real snow grain shape is more complicated and irregular (Kokhanovsky et al., 2005). We clarified in Line 68-74, 171-174 and 538-539 of the revised manuscript.

The hexagonal plate shape has been used as the default snow grain shape in the standalone version of SNICAR-AD v3 (Flanner et al., 2021). However, which snow grain shape should be prescribed in ELM still needs further investigation and analysis.

In the manuscript, the authors show that the impact of TOP on the results is quite large and larger than the impact of the mixing state. This is in my opinion an important result of this work and should get more attention. It should get mentioned in the abstract and the conclusions as one of

the main results. The impact of the mixing state of LAPs should not be overexaggerated in the abstract and conclusions as well.

We have documented the TOP parameterization and its impacts on the energy budget and snow processes in a previous study (Hao et al., 2021), so we have mainly investigated the interactions between the improved snow radiative transfer model and sub-grid topographic effects (TOP) in this work to minimize repetition. Indeed, apart from snow grain shape and mixing state of LAP-snow, topography also changes the apparent surface albedo and solar radiation absorbed by the surface, and in turn affects snow dynamics. The individual contributions of non-spherical snow shapes, mixing states of LAP-snow, and local topography to the change of snow and surface fluxes have different signs and magnitudes, and their combined effects may be negative or positive due to complex and non-linear interactions among the factors. We added the finding about the TOP effects in the abstract (Line 35-38), discussion (Line 464-474) and conclusions (Line 556-559) of the revised manuscript. We also changed the expression (e.g., avoid the use of such expression 'large impacts') to avoid overexaggerating the impacts of mixing state of LAP-snow.

**Specific comments**

P1 L18: "The mixing state of LAPs in snow also has large impacts…". In the manuscript you show that the impact of the mixing state is considerably smaller than compared to grain shape and TOP. So please rephrase.

We deleted 'large' in this sentence of the revised manuscript.

P1 L27: Please format units like $W/m^2$ with negative exponents, i.e., $W\ m^{-2}$, throughout the manuscript.

As suggested, we have reformatted the units throughout the manuscript.

P3 L42: Please provide the definition of albedo here as well.

We have replaced "albedo" with "broadband albedo", which is defined as the ratio of reflected to the incident radiative flux at the surface over the entire solar spectrum. We have added this definition in Line 45-46 of the revised manuscript.

P3 L44: 'Sky conditions' is unclear

Snow albedo is affected by the atmospheric conditions which determine the direct-to-diffuse ratio of incoming radiation. Thus we have replaced "sky conditions" by "atmospheric conditions" in Line 48 of the revised manuscript to be more clear.

P3 L67: Define the asymmetry factor.

Asymmetry factor represents the average cosine of the scattering phase angle and determines the fractions of backscattered and forward-scattered light. We have added this explanation in Line 159-160 of the revised manuscript.

P3 L69: Define the single-scattering albedo.

Single-scattering albedo describes the probability that a photon experiencing an extinction event is scattered as opposed to absorbed. We have added this explanation in Line 156-157 of the revised manuscript.

P4 L75: In my opinion, more should be said about the physical processes that make the albedo drop because of dry and wet deposition. So why is there a difference between the impact of dry deposition and wet deposition. Also, do you expect dry or wet deposition to be most relevant for the Tibetan Plateau?

We have revised this sentence as "The mixing state of LAPs in snow (i.e., external mixing and internal mixing) has large impacts on $\alpha_{sno}$" in Line 79-81 of the revised manuscript. The dry and wet depositions of LAPs can affect the mixing state of LAPs in snow and thus snow albedo. Such information is provided by the predefined aerosol deposition data or the atmospheric aerosol model. We clarified these in Line 215 of the revised manuscript.

P4 L102: haven't --> have not

Done.

P6 L140-141: " 1) SZA dependence of surface irradiance". This is a bit confusing, it suggests that you have implemented a new routine for SZA dependency, but if I understand it correctly, it is part of the SNICAR-AD model. Please clarify.

The original SNICAR-AD model assumes that clear-sky irradiance does not change with SZA. In the new version (i.e., SNICAR-AD v3), clear-sky irradiances are calculated for the full range of SZA (0–89° at 1° resolution) for each profile. We incorporated this new improvement in our ELM model. We clarified this in Line 151 of the revised manuscript.

P6 L144: There are six types of atmospheric profiles now included, but are clouds considered as well or is it only for clear-sky conditions? As clouds alter the spectral distribution of irradiance and limit direct radiation, it often has quite a large impact on the broadband albedo.

All of the six types of atmospheric profiles consider the clear-sky and cloudy conditions (see Table 1 in Flanner et al. (2021) for details). We clarified this in Line 152 of the revised manuscript. Spectral irradiances associated with clear-sky or cloudy atmospheres are selected internally in the SNICAR-AD model to match the specific conditions of direct or diffuse incident light.

P6 L147: Assuming that you will introduce the single-scattering albedo and asymmetry factor earlier, please also define the extinction cross section.

We replaced 'extinction cross section' with extinction efficiency. Extinction efficiency is the sum of scattering efficiency and absorption efficiency and represents the light attenuation ability of the particle. We added this explanation in Line 157-158 of the revised manuscript.

P6 L160: Why do you choose to investigate these three non-spherical grain shapes?

The three non-spherical shapes are typical snow grain shapes used to approximately represent the observed snow grains in the real snowpack (Dominé et al., 2003; Erbe et al., 2003; Liou et al., 2014), which capture the major morphological characteristics of observed snow grain structures. Moreover, the simulated snow albedo with the assumptions of the three non-spherical snow shapes showed better agreement with the observations (He et al., 2018). We clarified these in Line 68-74, 171-174 and 538-539 of the revised manuscript. Please see our response to the third question from Reviewer #2 for more details.

P7 L171: "… and Rs is the specific-projected-area-equivalent radius." This has already been said and can be removed here.

We have edited the text as suggested.

P7 L181: "…where $\omega_{dust}$ and $\omega_p$ are single-scattering albedo of pure snow and dirty snow…". I suppose it is vice versa? i.e., $\omega_p$ is for pure snow?

Yes. We have modified it.

P9 L228: A dot is missing after the citation.

We have fixed the citation.

Table 1: What impurity concentration do you assume?

We do not make any assumption about the impurity concentration. SNICAR-AD can prognostically simulate the evolution of the snow grain size and LAP concentration via deposition and redistribution. We clarified it in Line 133-134 of the revised manuscript.

P10 L237: $u$m --> µm

We have edited the text as suggested.

P10 L44: Why do you choose this simulation to be the control simulation? Is this the most realistic one? Please explain.

Generally, the use of "control" implies the original configuration, before any changes are implemented. The control simulation with the default settings in the original ELM is named ELM_Control, while the case with all the added parameterizations is named ELM_New (Table 1). In the revised manuscript, we compared both ELM_Control and ELM_New with MODIS snow data. We clarified this in Line 224-225 of the revised manuscript.

P10 L257-260: Could you elaborate a bit more on ANOVA? Also, what do you mean with 'maximum absolute relative difference' and 'mean absolute relative difference'?

ANOVA compares the means between different groups and determines whether they are statistically significantly different from each other. The maximum absolute difference is calculated as the maximum of the absolute value of the difference between two cases. The maximum relative difference is calculated as the maximum of the absolute value of the relative difference between two cases. The mean absolute difference is calculated as the mean of the

absolute value of the difference between two cases), and mean relative difference is calculated as the mean of the absolute value of the relative difference between two cases. We clarified these in Line 268-276 of the revised manuscript.

Fig. 2: For areas in the east, it is hard to see if there is a low snow cover or no snow at all. This can be solved by providing a separate color for no snow cover. Furthermore, The figure might be easier to read if discrete colors are used.

We masked the areas with low snow cover smaller than 0.01 as below. We also used the discrete colors for all spatial maps of the revised manuscript. We have also updated all other spatial maps to use discrete colormaps.

[Figure]

**Figure R4: Spatial distributions of $f_{sno}$ estimated from (a,e) ELM_Control, (b,f) ELM_New, (c,g) STC-MODSCAG/ STC-MODDRFS and (d,h) SPIReS for winter (a-d) and spring (e-h). The areas with $f_{sno}$ smaller than 0.01 are masked. This figure corresponds to Figure S2 in the revised manuscript.**

Fig 3 and all other boxplots. Please also state that what is on the x axis; i.e., that fsno and SAR as a function of elevation are shown.

We have made the suggested change.

Fig. 4: Same as Fig 2. Also, 'Same as Figure 3' is written, while it is the same as Fig. 2

We have made the suggested change.

P11 L277: "… although their spatial patterns are similar." This is somewhat hard to see in the figures. A figure with the difference between ELM and the observations could help with that.

We added such a figure as Figure R1 in the revised manuscript. This figure shows that ELM_Control overall overestimates snow cover fraction for winter, but underestimates snow cover fraction in spring. ELM_New reduces the bias of snow cover fraction in spring compared to ELM_Control.

P12 L290: "The difference may be due to the overestimation of snow grain size…". Can you explain a bit more why this may be the case? As grain size has a strong impact on the albedo, modelling this incorrectly could lead to large differences with observations, potentially overshadowing any grain shape or mixing state effects. It seems to me like it may have a large impact on the results here and should be mentioned.

Indeed, snow grain size has large impacts on the snow albedo. Figure R5 shows the spatial patterns of snow grain size of ELM and MODIS data. Clearly ELM overestimates snow grain size compared to MODIS data, which is also identical to the results in (Sarangi et al., 2020). The overestimation of snow grain size in ELM may account for the difference between MODIS and ELM. Besides, both the impacts of non-spherical snow grain shape and mixing states are sensitive to snow grain size (He et al., 2018a). However, there are large uncertainties in modeling the snow aging processes to represent the evolution of snow grain size in ELM (Figure R5), which needs further improvements. We added this figure and the corresponding analysis in Line 302-303 and 462-463 of the revised manuscript.

[Figure]

**Figure R5: Spatial distributions of snow grain size estimated from: (a,e) ELM_Control, (b,f) ELM_New, (c,g) STC-MODSCAG/STC-MODDRFS and (d,h) SPIReS for winter (a-c) and spring (d-f). The areas with $f_{sno}$ smaller than 0.01 are masked. This figure corresponds to Figure S3 in the revised manuscript.**

P14 L316: "Western regions show larger RFs induced by all LAPs than the eastern regions". Looking at Fig. 2 there is almost no snow cover in the east, so logically barely any RF changes are visible there.

We masked the areas with low snow cover smaller than 0.01 to distinguish the snow-free regions. We also used the discrete colors for all spatial maps of the revised manuscript.

P14 L324: I assume you mean Table 3, not Table 1?

Modified.

P14 L328: "… which is identical to" --> which is similar to.

We have modified the text as suggested.

P15 L341: Are all shown changes in Fig. 6 significant? If not, please also illustrate significance on the maps.

Yes, all the differences in the six sub-figures of Fig. 6 are significant based on ANOVA (p<0.05). We clarified this point in Line 357-358 of the revised manuscript.

Caption of Fig. S4 and S5: I suppose you mean $\alpha_{sno}$ instead of $A_{sno}$?

We have modified the text as suggested.

Fig. 7: A bit confusing that the color scale is now inverted compared to Fig. 6. To help solve this confusion, please explicitly state in the manuscript that the differences are now negative.

We explicitly clarified in the captions of both Figures 6 and 7 in the revised manuscript.

P19 L389: "… has larger effects on $F_{lat}$ than $F_{sen}$". Why?

Snow grain shape affects snow albedo and thus surface albedo, which further affects net solar radiation and thus surface energy balance. We clarified this mechanism in Line 406-409 of the revised manuscript. The non-spherical impacts on latent heat and sensible heat fluxes vary with space and are related to the relative magnitude of latent heat and sensible heat fluxes. We deleted this sentence to avoid misunderstanding.

P19 L391 - 392: "The differences in … than those for Koch snowflake shape". Can you explain why?. Similarly on P19 L395 – 396, please explain why you would expect a sign change.

Snow grain shape affects the differences in the energy balance terms between the internal and external mixing of LAP-snow, because non-spherical grains tend to have larger $\alpha_{sno}$ and could affect SAR induced by LAPs. We have rephrased the first sentence to clearly show the results and the corresponding explanation in Line 357-359 of the revised manuscript.

The mixing state effects on surface energy balance have an opposite sign compared to the non-spherical shape effects, because internal mixing leads to smaller snow albedo than external

mixing, while non-spherical grain shape has large snow albedo than spherical grain shape. We added these explanations in Line 378-379 of the revised manuscript.

P21 L417 – 418: One of the figure references can be omitted.

We have deleted the first figure reference.

P23 L455 – 456: "… to the reported values in the existing study". What do you mean? It is not clear to me what the authors want to say here.

We have deleted this paragraph, considering that this part is not well related to this study.

P23 L459 – P24 L466: "However, different studies … or annual scales". I am not sure if I understand what you want to say with this part.

We have deleted this paragraph, considering that this part is not well related to this study.

P24 L470: "Mixing state of LAP-snow also has large impacts on SAR". Looking at the results of this manuscript, it looks like the impact is considerable smaller than the impact of grain shape and TOP. Please rephrase.

We have deleted 'large' in the manuscript.

**References**

Dominé, F., Lauzier, T., Cabanes, A., Legagneux, L., Kuhs, W. F., Techmer, K., and Heinrichs, T.: Snow metamorphism as revealed by scanning electron microscopy, Microscopy Research and Technique, 62, 33-48, 2003.

Erbe, E. F., Rango, A., Foster, J., Josberger, E. G., Pooley, C., and Wergin, W. P.: Collecting, shipping, storing, and imaging snow crystals and ice grains with low-temperature scanning electron microscopy, Microscopy research and technique, 62, 19-32, 2003.

Flanner, M. G., Arnheim, J. B., Cook, J. M., Dang, C., He, C., Huang, X., Singh, D., Skiles, S. M., Whicker, C. A., and Zender, C. S.: SNICAR-ADv3: a community tool for modeling spectral snow albedo, Geosci. Model Dev., 14, 7673–7704, https://doi.org/10.5194/gmd-14-7673-2021, 2021.

He, C., Flanner, M. G., Chen, F., Barlage, M., Liou, K. N., Kang, S., Ming, J., and Qian, Y.: Black carbon-induced snow albedo reduction over the Tibetan Plateau: uncertainties from snow grain shape and aerosol–snow mixing state based on an updated SNICAR model, Atmos. Chem. Phys., 18, 11507-11527, 2018.

Liou, K. N., Takano, Y., He, C., Yang, P., Leung, L. R., Gu, Y., and Lee, W. L.: Stochastic parameterization for light absorption by internally mixed BC/dust in snow grains for application to climate models, Journal of Geophysical Research: Atmospheres, 119, 7616-7632, 2014.

---

## Author Comment (AC3)

**Reviewer 3:**

This study incorporates several new features into the representation of snow albedo in the E3SM model, including non-spherical snow grains, internally-mixed dust and black carbon within ice grains, and sub-grid topographic effects. The study assesses the impacts of these improvements on the simulated snow cover and surface energy budget, including impacts of individual effects as well as the combined effects of all model changes operating simultaneously. Overall, the paper is well-organized and very well-written. I have only a few minor comments, and I recommend that the manuscript be published after these are addressed.

Thank you for these useful comments and suggestions. We have revised the manuscript carefully.

**Minor comments:**

Section 4.3: In the sensitivity studies that apply internal mixing of LAPs, are all of the particles assumed to be internally-mixed, or only the proportion of deposited aerosol that was simulated to be internally-mixed? (And is that proportion actually simulated? My understanding is that the MAM aerosol model simulates the mixing state of aerosols, so that information could, in principle, be utilized, but it is not clear that such information is being extracted and utilized in the model experiments). Either way, please expand the discussion on this issue, including any implications for the magnitude of impact assessed in the sensitivity studies.

In this study, we performed sensitivity analyses using offline ELM simulations in which predefined aerosol dataset for dust and black carbon (BC) was used. The dust particles in ELM are assumed to be either internally-mixed (Sph_BCInt_DExt_PP) or externally-mixed with snow (Sph_BCExt_DExt_PP). The aerosol dataset includes deposition rate of both hydrophobic BC and hydrophilic BC. This study assumes that hydrophobic BC is all externally mixed with snow, while hydrophilic BC can be all externally-mixed (Koc_BCExt_DExt_PP) with snow or internally-mixed (Sph_BCExt_DInt_PP) with snow. We have clarified these details in Line 209-227 of the revised manuscript.

Indeed, in reality, both the snow grain shape and mixing state of LAP-snow are spatially-inhomogeneous and time-varying (Räisänen et al., 2015). For instance, dust may be partially internally and externally mixed with snow grains and simply assuming external mixing of dust-snow could underestimate the dust effects, while assuming fully dust-snow internal mixing will overestimate the dust effects (Shi et al., 2021). We added these discussions in Line 538-542 of the revised manuscript.

Currently, the four-mode version of Modal Aerosol Module (MAM4) assumes the internal mixing of aerosol particles (BC and dust) in the same size bin and external mixing between different size bins within the atmosphere. For the interactive atmosphere-land simulations using the current E3SM, the mixing state of aerosols in MAM4 is used to inform the partitioning of hydrophobic and hydrophilic BC in the deposition fluxes to land surface. However, the mixing between BC and dust components, BC-snow, or BC-dust is not considered. Besides, land-atmosphere interaction is neglected by performing offline ELM simulations in this study. Further investigation is needed to couple the atmosphere model with the MAM scheme and ELM to share more information about the mixing state of LAP-snow and LAP-LAP. We stated these limitations in Line 542-543 of the revised manuscript.

line 115: "... control ELM simulation with the default settings..." - Please clarify whether the default settings are the old or new default settings, i.e. with or without all of the changes described in this study. The use of "control" would generally imply the original configuration, before cay changes are implemented.

The control simulation with the default settings in the original ELM is named ELM_Control, while the case with all the added parameterizations is named ELM_New (Table 1). We clarified this in Line 224-225 of the revised manuscript.

Line 318: "... because LAP-induced SAR is larger in spring ..." - While the model SAR is definitely larger in spring than winter, it is not totally clear from Figs 3 and 4 that the *observed* SAR is larger in spring. (And I am surprised that it is not, given the likelihood of melt-induced

surface accumulation during spring and higher frequency of springtime dust storms in this region). Please comment on this.

We have clarified that spring has larger LAP-induced SAR in the ELM simulations to avoid misunderstanding in Line 334-335 of the revised manuscript. We also stated the difference between the model-based and MODIS-based results and discussed the uncertainties of both model simulations and MODIS retrievals. The model simulations depend on the prescribed aerosol deposition data, while the MODIS retrievals depend on the quality of observed reflectance and algorithm assumptions. Only limited field measurements showed that the snowpacks in the Indus Basin are clean in winter (Negi et al., 2010). More field measurements over the TP are needed to evaluate the model simulations and MODIS retrievals to advance our understanding of the snow darkening effects of LAPs on snow and their spatial and seasonal variations. The ongoing and upcoming hyperspectral satellites such as PRISMA (PRecursore IperSpettrale della Missione Applicativa) mission, the Surface Biology and Geology (SBG) led by NASA and the Copernicus Hyperspectral Imaging Mission for the Environment (CHIME) led by European Space Agency (ESA) will be promising for improving the remotely sensed estimates of LAP-induced SAR. Long-term field measurements of snow grain characteristics, $\alpha_{sno}$ and LAP concentrations over the TP are needed to evaluate the model simulations and advance our understanding of snow grain shape and mixing state effects. We added these discussions in Line 509-533 of the revised manuscript.

Line 394: "Overall, the effects of mixing state of dust-snow are smaller than the effects of mixing state of BC-snow." - Why? Please explain.

We found that both the effects of the mixing state of dust-snow and BC-snow vary with space (Figure 12 in the revised manuscript). In fact, the effects of mixing state of LAP-snow depend on the specific LAP concentration for the target pixel. This sentence may mislead the readers and thus we have deleted it in the revised manuscript.

Line 511: "... but the real snow grain shape may be more complicated and irregular..." - I think you can safely say that the real snow grain shape \*is\* more complicated and irregular.

Agree. Done.

**References**

Negi, H. S., Singh, S. K., Kulkarni, A. V., & Semwal, B. S. (2010). Field-based spectral reflectance measurements of seasonal snow cover in the Indian Himalaya. International Journal of Remote Sensing, 31(9), 2393-2417.

Shi, T., Cui, J., Chen, Y., Zhou, Y., Pu, W., Xu, X., Chen, Q., Zhang, X., and Wang, X.: Enhanced light absorption and reduced snow albedo due to internally mixed mineral dust in grains of snow, Atmos. Chem. Phys., 21, 6035-6051, 2021.

Räisänen, P., Kokhanovsky, A., Guyot, G., Jourdan, O., and Nousiainen, T.: Parameterization of single-scattering properties of snow, The Cryosphere, 9, 1277-1301, 2015.

---

## Author Comment (AC5)

Dear Editors and Reviewers,

Thank you very much for your detailed and constructive comments and suggestions for our manuscript. We found the comments very helpful for improving our manuscript and have revised the manuscript accordingly.

Below are the major revisions that we have made:

1. We have used new versions of MODIS data (i.e., STC-MODSCAG/STC-MODDRFS and SPIReS) as the benchmarking datasets covering the entire Tibetan Plateau. Following the suggestion of Reviewers 1 and 2, we have added a comparison between the original (ELM_Control) and new versions (ELM_New) of ELM and MODIS data.
2. We have performed additional analysis of the impacts of snow grain shape and mixing state of LAP-snow on energy budget and water cycle and stressed the importance of topographic effects.
3. We have added more details about the variable definitions, model assumptions on snow grain size and LAP concentrations, statistical methods and metrics, remote sensing data, experimental configurations, and mechanistic explanations on the results.
4. We have improved the figure quality by using discrete colors and different colors for the pixels with no snow cover and modified the figure captions with more details.

Timbo Stillinger has put in a lot of effort to generate the new version of SPIReS dataset and helped improve the manuscript, so we have added him as a coauthor; all coauthors agreed to this addition. The itemized responses to the comments and suggestions from the editors and reviewers are provided below in blue font. We hope our responses address the concerns raised in the last round of review and look forward to your decision on the publication of our manuscript.

Best regards,

Dalei Hao (on the behalf of all authors)

/

---

## Author Response (AR2)

**Comments to the author**:

The reviewers are clearly very enthusiastic about the manuscript. GMD has a potential audience with broad interests and I have a couple of comments to make as a generalist climate scientist who is not an expert in snow/albedo modelling.

We appreciate your comments and suggestions, and we have revised the manuscript accordingly.

I am a little disappointed that while implementing new features into an ESM, you only test the results on one part of the globe. It seems very dangerous. Can you comment on this? Have you looked at other regions? Does it all go horribly wrong, or did you choose the Tibetan plateau because actually it highlights the differences better?

We selected the Tibetan Plateau (TP) as a testbed in the study, based on several reasons:

1) As a hotspot region under climate change, TP is characterized by complex topography and frequent snow cover. TP is also characterized by a thin snowpack, making its albedo effects particularly susceptible to climate warming (Liu et al., 2000). Besides, the radiative impact of light absorbing particles (LAP) (e.g., black carbon and dust) may be most climatically relevant in the TP (Skiles et al., 2018). Thus, TP is an ideal region for us to analyze and compare the impacts of snow grain shape, mixing state of LAP in snow, and topography. We presented these in Section 3.1.
2) Two MODIS snow property products (i.e., STC-MODSCAG/STC-MODDRFS and SPIReS) provide good references for evaluating our model. However, unfortunately, both products are not globally available yet and only include few typical regions (e.g., TP).
3) There are many previous studies that have studied the radiative forcing of LAP over the TP, which make it possible to compare our results with the existing studies. We presented the comparison in Section 4.2.

We are extending our study to other regions. In a follow up study that is currently under review in *The Cryosphere* ( Hao et al., 2022; https://egusphere.copernicus.org/preprints/2022/egusphere-2022-1097/), we have applied the newly developed snow albedo model to Western US for evaluating simulated snow processes in E3SM land model. Further applications and evaluation over the Arctic and Antarctic would be a good future direction, as more and more field measurements and remote sensing data become available.

Besides, we are currently conducting land-atmosphere coupled experiments with the new snow albedo model to analyze the LAP impacts when considering the land-atmosphere feedback. Thus, we stated the limitations of this study and stressed future extended study at a global scale with the consideration of land-atmosphere interactions in Line 544-546 of the revised manuscript.

In terms of overall context, I need another anchor. You clearly state in the abstract that the inclusion of internal mixing of dust (and other stuff) is a smaller effect than changing the shape of the snow

grains, but what about the total influence of dust (and other stuff)? Am I right to be thinking that this is, in practice, is a much bigger effect than changing snow grain shape? Please add a sentence to the manuscript to clarify.

Good point! Indeed, the impacts of LAP on snow may be comparable to or even larger than the impacts of snow grain shape, which depends on the LAP concentration in snow. Figure 5(e-f) shows that the snow albedo reduction induced by LAP can be close to 0.1 over the western TP. Figure 7(a-c) shows the difference in snow albedo between non-spherical and spherical grain shapes. Comparing Figures 5(e-f) and 7(a-c) shows that the impacts of LAP can be comparable to or even larger than the impacts of non-spherical shape in spring, especially for the western TP where the LAP concentration in snow is high. Similar conclusion can be drawn from He et al. (2018).We have now added such discussion in Line 460-465 of the revised manuscript.

The last sentence of your author contributions paragraph can be removed. Or maybe I have misunderstood what you are trying to say. All 12 are accounted for in by name in the preceding sentences where it also states that they all edited the manuscript. I think we always assume that authors improve the work even if they do so by disagreeing with some of the content. The author contributions paragraph is especially useful for highly collaborative sciences such as climate science, as it enables all authors to accept responsibility for their own contributions, even though they do not fully comprehend all the details of the whole manuscript.

We agree with you. We have deleted the last sentence in the section of author contributions.

**References**

Hao, D., Bisht, G., Rittger, K., Stillinger, T., Bair, E., Gu, Y., and Leung, L. R.: Evaluation of snow processes over the Western United States in E3SM land model, EGUsphere [preprint], https://doi.org/10.5194/egusphere-2022-1097, 2022.

He, C., Flanner, M. G., Chen, F., Barlage, M., Liou, K.-N., Kang, S., Ming, J., and Qian, Y.: Black carbon-induced snow albedo reduction over the Tibetan Plateau: uncertainties from snow grain shape and aerosol–snow mixing state based on an updated SNICAR model, Atmos. Chem. Phys., 18, 11507–11527, https://doi.org/10.5194/acp-18-11507-2018, 2018.

Liu, Xiaodong, and Baode Chen. "Climatic warming in the Tibetan Plateau during recent decades." International Journal of Climatology: A Journal of the Royal Meteorological Society 20, no. 14 (2000): 1729-1742.

Skiles, S.M., Flanner, M., Cook, J.M. et al. Radiative forcing by light-absorbing particles in snow. Nature Clim Change 8, 964–971 (2018). https://doi.org/10.1038/s41558-018-0296-5.

---

## Author Response (AR3)

**Comments to the author**:
Sorry to come back with another thing, but I only just found the supplement. I'd been wondering why there was no figure showing the shapes, and I see it is sitting in the supplement! Some of the supplemental figures are referenced several times in the manuscript, so they should be included in the main manuscript. For those that are not referenced in the main manuscript, there should be a paragraph of text in the supplement explaining what the figures are. In short, the supplement should contain information that is supplemental - of interest to readers, but not critical to the paper. Note that supplements are not archived so the same level as the main paper, so anything at all critical needs to be in the main manuscript.

We appreciate your suggestions. We agree that the snow shape figure is important one, and thus we have moved it the main text from the supplementary materials. Considering that the other figures in the supplementary materials are not critical to the main points and conclusions of this paper and just for the additional interests of the readers, all of them are kept in the supplementary materials and cited in the main text. We have also adjusted the figure number accordingly throughout the revised manuscript.

Thanks a lot for your nice review!

---

## Author Response (AR4)

**Comments to the author**:

Thanks for including the shapes figure. Now I feel much less of a fool for not instinctively knowing what a Koch snowflake is like! :-) I would still like to see an introductory paragraph in the supplement, but I leave that as a technical correction for you to consider, and am otherwise happy to accept the paper.

We appreciate your suggestions. The Koch snowflake, also known as the Koch star or Koch island, is a fractal curve (Koch 1904). As shown in Figure 1, it is built by starting with an equilateral triangle, removing the inner third of each side, building another equilateral triangle at the location where the side was removed, and then repeating the process indefinitely. We have added a simple explanation of the Koch snowflake in Line 172 of the revised manuscript.

References

Koch, H.: Sur une courbe continue sans tangente, obtenue par une construction géométrique élémentaire, Arkiv for Matematik, Astronomi och Fysik, 1, 681-704, 1904.

Thanks a lot for your nice review!